# METAMUSE: ALGORITHM GENERATION VIA CREATIVE IDEATION

**Ruiying Ma**
University of California, Berkeley

**Chieh-Jan Mike Liang**
Microsoft Research

**Yanjie Gao**
Microsoft Research

**Francis Y. Yan**
University of Illinois Urbana-Champaign

## ABSTRACT

Designing system algorithms remains challenging, where the discontinuous nature of the solution space often forces system engineers to rely on generic heuristics at the expense of performance. We study whether LLMs can practically drive algorithm generation, and find that they are biased towards well-known generic designs, rather than making the creative leaps needed to navigate the discontinuous solution space. To address this limitation, we introduce MetaMuse, a framework for creative ideation built on three self-reflection principles: *(1)* quantifying solution diversity and usefulness in measurable performance space, rather than abstract idea space, *(2)* steering ideation through external stimuli, rather than internal randomness, and *(3)* constructing executable solutions using waypoint reasoning, rather than free-form chain-of-thought. Considering two critical online problems at a global cloud provider, extensive evaluations show that MetaMuse can generate high-performing solutions: it reduces cache misses by up to 35.76% in cache replacement and reduces bin usage by up to 30.93% in online bin packing.

## 1  INTRODUCTION

Designing system algorithms continues to be a central challenge in computing systems. Traditionally, the development of such algorithms has been a manual and labor-intensive process. Our experience at a global cloud provider indicates that even seemingly simple algorithms used in production—such as cache replacement for data storage or bin packing for job scheduling—can require tens of thousands of engineering hours to design. As a result, practitioners often resort to generic heuristics from the literature, e.g., least-recently used (LRU) and least-frequently used (LFU) for cache replacement, and first-fit for bin packing, which frequently result in suboptimal performance.

This paper asks whether large language models (LLMs) can practically drive **algorithm generation**, *with an emphasis on principles to transform this task into a systematic process*. The core challenge in system algorithm design arises from the nature of its solution space: it is an inherently discontinuous space, where even a small change in algorithm design (e.g., data structure or control flow) can lead to sharp and non-linear changes in performance. Although it is sometimes possible to estimate the upper-bound performance, searching for practical solutions that approach this bound remains non-trivial. Furthermore, the discontinuous solution space does not provide sufficiently predictable patterns or a smooth landscape to guide the search.

Due to this discontinuity, we approach the algorithm generation task from a different angle, framing it as a sampling process in the solution space where the LLM attempts to generate distinct solutions at each step. This generative process represents a sequence of leaps across the discontinuous solution space (Bubeck et al., 2023), which we formulate as **creative ideation** for LLMs. In fact, the systems community has long hypothesized algorithm design as a discovery process of ideas (Kant, 1985).

To study the algorithm generation task, we focus on high-impact problems at a global cloud provider: cache replacement and online bin packing. Our initial attempts of repeatedly sampling GPT-4o, Llama3.3-70B, and DeepSeek-V3 show that LLMs are fundamentally hindered by **availability bias** (Tversky & Kahneman, 1973)—LLMs are trained to output the most likely sequence of words, according to training datasets. As a result, solutions tend to cluster around well-known heuristics in the literature, e.g., LRU and LFU for caching. Furthermore, we find that this bias cannot be practically addressed through LLM hyperparameters like temperature (Ackley et al., 1985).

---

This work was done when Ruiying Ma was an intern at Microsoft Research.

The key to creative ideation is exploiting knowledge that LLMs assume to be probabilistically irrelevant to the given problem. What is missing is a self-reflection process, which thinks how to generate subsequent solutions by inspecting what solutions have been generated so far. In realizing such a self-reflection framework, MetaMuse, we see three model-agnostic principles surfacing to best guide this self-reflection. First, evaluating diversity of generated solutions should be grounded in the measurable *feedback space* (e.g., simulation performance of system algorithms), rather than the abstract idea space (Sankar & Sen, 2024). Second, steering the ideation is achieved through *external stimuli* (e.g., keywords), rather than internal randomness (Chen & Ding, 2023). Third, developing executable solutions from external stimuli takes structured checkpoint-based steps, or *waypoint reasoning*, rather than free-form chain-of-thought (Wei et al., 2023; Mehrotra et al., 2024).

Our work makes the following contributions:

1. Empirical analysis of LLMs' fundamental limitations in the algorithm generation task (§2). We shed light on the impracticality of relying on LLMs' internal randomness for ideation, based on two online problems at a global cloud provider: cache replacement and online bin packing.

2. Principled framework for creative ideation, MetaMuse (§3). We systematically combine three self-reflection principles to guide the discontinuous leaps necessary to overcome LLMs' bias.

3. Practical evaluation of the online algorithm generation task (§4). We show that *(1)* MetaMuse generates better cache replacement and bin packing solutions, outperforming LLM-based baselines (by up to 9.89% fewer cache misses, and up to 21.06% less bin usage) and human heuristics (by up to 35.76% fewer cache misses, and up to 30.93% less bin usage); *(2)* MetaMuse has up to $1.78\times$ more distinct cache replacement solutions and $1.80\times$ more distinct bin packing solutions, than LLM-based baselines; *(3)* MetaMuse has a low per-solution cost, up to 2.16 cents with GPT-4o.

## 2 BACKGROUND AND MOTIVATION

### 2.1 SYSTEM ALGORITHM DESIGN

System algorithms define how computing systems behave. They are typically designed to optimize some performance objectives (e.g., cache hit ratio), for some scenarios or workloads (e.g., web servers). The difficulty of designing such algorithms arises from the nature of their solution space. It is discontinuous, where even a small change in algorithm design (e.g., the use of data structures) can lead to sharp and non-linear changes in performance. In addition, the discontinuity does not provide sufficiently predictable patterns or a smooth landscape to guide the search. This is a significant departure from prior efforts on auto-tuning system config parameters (Alipourfard et al., 2017; Cortez et al., 2017; Liang et al., 2020), which can be formulated as numerical optimization in most cases.

To this end, our work explores the use of LLMs to design heuristic algorithms, i.e., the algorithm generation task. Due to the discontinuous solution space, we formulate this task as a sampling process over the solution space, in which LLMs aim to generate distinct solutions at each step. Conceptually, solving such a discontinuous task requires certain "Eureka" ideas that constitute leaps towards the final solution (Bubeck et al., 2023). Here, we refer to these leaps as *creative ideation*.

### 2.2 CREATIVE IDEATION

The goal of creative ideation is to discover useful solutions to a user-given problem, through a process of generating diverse solutions over time. The first requirement is *usefulness*—a generated solution should be relevant to the problem. The second requirement is *diversity*—a generated solution should produce feedback (or an outcome) that is unseen in the current process (Boden, 1998). As we accumulate diverse and useful solutions, the process advances towards finding the optimal solution.

For the algorithm generation task, creative ideation can be formulated as the following iterative process. Each iteration takes in the problem statement of the algorithm, along with the set of functions to be implemented. For example, most caching algorithms can be abstracted into `insert` and `evict` functions (Yang et al., 2020). Optionally, we can also incorporate metadata and feedback from previously generated solutions. At iteration $i$, the output is an executable cache solution, $c_i$, with all functions coded. We can optionally provide feedback to iteration $i + 1$, by evaluating the hit ratios of previous solutions $(c_1^{hit}, ..., c_i^{hit})$, on a user-given workload trace in an environment such as simulators. After $n$ cache solutions, we select the best-performing one from $(c_1, ..., c_n)$.

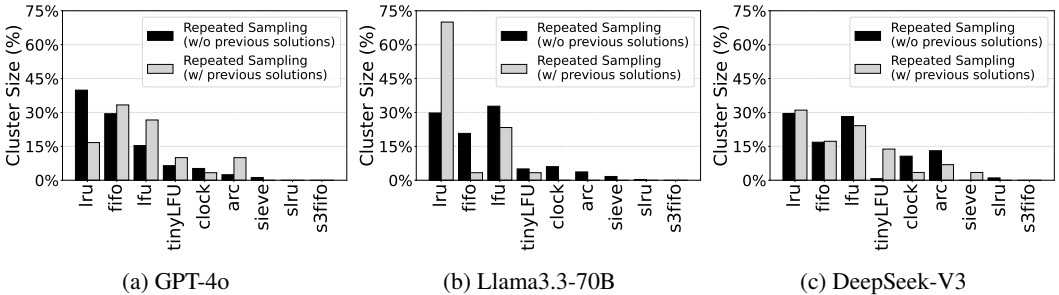

Figure 1: Repeatedly sampling LLMs can lead to biased solutions. In cache replacement, solutions tend to cluster around well-known heuristics in the literature, such as LRU, LFU, and FIFO.

### 2.3 LLM LIMITATIONS IN CREATIVE IDEATION

Our work is motivated by the observation that the creative ideation capability of LLMs is fundamentally constrained by the very mechanism that enables it—next-token prediction. In cognitive terms, responses are shaped by what has been frequently or recently seen, a phenomenon known as *availability bias*. For LLMs, this bias stems from the frequency distribution of data seen in training.

**Impacts on algorithm generation.** Due to this bias, LLMs tend to generate certain classical designs. We illustrate this phenomenon in the context of cache replacement, using GPT-4o (version: `1120`), Llama3.3-70B (version: `Instruct`), and DeepSeek-V3 (version: `0324`).

Figure 1 shows our attempts with *Repeated Sampling* (Brown et al., 2024; Snell et al., 2024). At iteration $i$, each LLM is prompted to generate a new algorithm design $c_i$ using a temperature of 1. To isolate creativity from coding ability, $c_i$ is then implemented in Python by the same model, GPT-4o. The resulting implementation is evaluated in a simulator to measure hit ratios on 30 synthetic traces. We examine two variants of Repeated Sampling that differ in whether the prompt at each iteration includes all previously generated solutions (i.e., solution descriptions and implementations). Figure 1 clusters 1,200 solutions by assigning them to the nearest centroids of established human heuristics, based on the similarity of their hit-ratio vectors. We observe that LLM-generated solutions concentrate around well-known heuristics, such as LRU, LFU, and first-in-first-out (FIFO). Moreover, making Repeated Sampling aware of previous solutions does not mitigate this bias.

Even if LLMs are instructed to generate new solutions by mutating previous solutions (Novikov et al., 2025; Sharma, 2025), we observe a bias towards tweaking the solution's scoring function. LLMs tend to ignore other design dimensions such as data structure, hierarchical architecture, etc.

**Is adjusting LLM hyperparameters all you need?** One such parameter is temperature (Ackley et al., 1985), which internally regulates randomness in the generative process. Temperature smooths the probability distribution of next-token candidates, as computed by the softmax of their logits. At higher temperatures, high probabilities are decreased, and low probabilities are increased.

Unfortunately, high temperatures only marginally mitigate availability bias without fundamentally addressing it. The reason is that increasing temperature smooths the probability distribution, which is a monotonic transformation retaining the relative ranking of output token candidates. Even at extreme temperatures, the distribution would approach uniformity, resulting in incoherent LLM outputs.

### 2.4 RELATED WORK

The first category is automatic heuristic design (Liu et al., 2024b; Ye et al., 2024; Zheng et al., 2025; Romera-Paredes et al., 2024; Novikov et al., 2025; Sharma, 2025; Dwivedula et al., 2025; Lange et al., 2025; Cheng et al., 2025). These efforts mostly rely on LLMs to evolve an initial population of candidate designs through mutations and crossovers. This dependence on the initial population inherently constrains the reachable solution space. In addition, availability bias can influence how the population evolves. Our case study reveals a bias towards frequently tweaking the cache algorithm's scoring function, rather than leaping to other design dimensions. MetaMuse addresses these limitations by making leaps in the solution space.

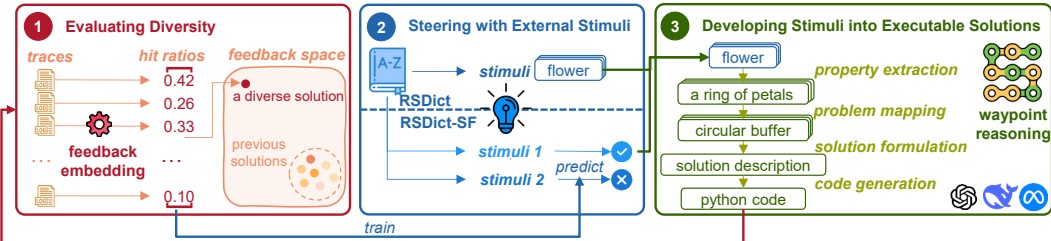

Figure 2: MetaMuse reflects on previously generated solutions to inform the creation of subsequent ones. Each iteration goes through three steps: evaluating the diversity of existing solutions (§3.1), steering ideation through external stimuli (§3.2), and developing executable solutions (§3.3).

There are also efforts exploring human-LLM collaborations to produce diverse outputs (Liu et al., 2024c; Wan et al., 2024; Vaccaro et al., 2024). However, they require human involvement and can be subject to human biases. In contrast, MetaMuse posits that LLMs themselves are capable of thinking "outside the box" (Mehrotra et al., 2024), and proposes combining three self-reflection principles to achieve this capability (§3).

# 3 METAMUSE

In this work, we introduce *MetaMuse*, a creative ideation framework that addresses availability bias through principled self-reflection. The key is to enable LLMs to think "outside the box," leveraging knowledge that otherwise seems probabilistically irrelevant to the user's problem.

Figure 2 illustrates a single iteration of MetaMuse, which outputs one executable solution. The first step evaluates the diversity of solutions generated so far (§3.1) based on embedding vectors constructed from each solution's feedback (e.g., cache hit ratios, as shown in the figure). The second step steers ideation by selecting a set of stimuli (e.g., the keyword "flower" in the figure). §3.2 describes two stimuli selection strategies in the current implementation: RSDict and RSDict-SF. Finally, MetaMuse develops the stimuli into executable solutions (§3.3), through four waypoints of reasoning: property extraction, problem mapping, solution formulation, and code generation.

## 3.1 EVALUATING DIVERSITY

MetaMuse grounds diversity evaluation in feedback space rather than idea space. In our case study, the former refers to performance measurements such as cache hit ratios, whereas the latter refers to semantic embeddings of algorithm descriptions or code implementations. MetaMuse represents each solution using a **feedback embedding**, which is a vector $[p_1, p_2, \ldots, p_n]$ containing performance measurements across $n$ workload traces. Ideally, these traces should be diverse, and one approach is to randomly synthesize them using tools from the systems community (e.g., libCacheSim (Yang et al., 2020) for cache workloads).

This principle is motivated by the observation that semantic differences do not necessarily imply distinct solutions. Considering a case where GPT-4o generated functionally equivalent LFU-based solutions twice from different design descriptions: "*The cache evicts the object that is least frequently accessed,*" and "*The cache evicts the object with the lowest priority. Upon a cache hit, the object's priority is incremented.*" Code semantics exhibit similar shortcomings, where a solution implemented with a linked list or a priority queue can yield the same hit ratio.

Furthermore, feedback embeddings have two desirable properties. First, because each dimension is a quantitative metric conveying magnitude, the Euclidean distance between two embeddings is a direct indication of how different the corresponding solutions are. In the LFU example above, those two solutions would be considered equivalent as their distance is 0. Second, feedback metrics typically lie within a fixed range, either by definition or through normalization (e.g., 0–100% for cache hit ratio). As a result, the $n$-dimensional embedding space is inherently structured and bounded, allowing us to compute a well-defined steering direction (§3.2).

## 3.2 Steering with External Stimuli

The objective of steering is to guide LLMs to generate solutions, targeting regions of the feedback embedding space. Rather than relying on the model's internal randomness, MetaMuse introduces **external stimuli** as the starting point for ideation. These stimuli may be unrelated to the problem domain, forcing LLMs to associate with knowledge that appears probabilistically irrelevant. One domain-agnostic instantiation of such stimuli is a set of keywords from an English dictionary.

Realizing this principle of steering presents practical challenges. The first is the source of stimuli. In the case of keywords, while a large dictionary provides broad keyword coverage, it also includes many keywords of little value. An example is technical jargon, e.g., "prolamin" in biology. In our case study, if LLMs are forced to associate these keywords, they would frequently use them as variable names, rather than as inspiration to ideate new solutions. The second challenge is selecting $s$ stimuli at each iteration. §3.2.1 next presents selection strategies in our current implementation.

### 3.2.1 Stimuli Selection Strategies

Our current implementation of MetaMuse includes two strategies: RSDict and RSDict-SF.

**RSDict.** RSDict is a stateless strategy that randomly selects $s$ keywords from the dictionary. Since RSDict does not rely on the evaluation results of previous solutions, it is applicable in settings where evaluation is infeasible or costly (e.g., when human judges are required).

**RSDict-SF.** Based on RSDict, RSDict-SF further relies on self-feedback, leveraging the feedback embeddings of prior solutions to compute a steering direction. The steering direction is specified as a target feedback embedding, which serves as the objective that RSDict-SF should aim to achieve in subsequent solutions. In our case study, this mechanism supports both exploration (for diversity) and exploitation (for usefulness). For exploration, the steering direction is defined as the point in the embedding space that is farthest from all previous solutions, i.e., the embedding with the greatest Euclidean distance from the existing set. For exploitation, we define the target embedding by setting all dimensions to a high value (e.g., 100% cache hit ratios across all traces).

Given a steering-direction embedding, RSDict-SF computes a set of $s$ stimuli that are likely to develop into solutions close to the target. Our current implementation formulates this step as a prediction problem and solves it with Gaussian Process Regression (GPR). Specifically, given $s$ stimuli, we use their corresponding *observations* to predict the solution's feedback embedding dimension by dimension, without generating the solution itself. Each observation is a natural-language sentence generated by LLMs after mapping one stimulus to the problem domain; we describe this process in detail in §3.3. We first compute the *semantic embeddings* (e.g., by SBERT Reimers et al. (2019)) of the $s$ observations, and use $n$ GPR models, $(\mathcal{M}_1, \ldots, \mathcal{M}_n)$, to predict each of the $n$ dimensions in feedback embedding. These models allow RSDict-SF to predict the expected feedback embedding for any set of stimuli without generating the solution. The GPR models are refitted at each iteration with all previous solutions. We adopt a dot-product kernel for the GPR models. This kernel $\mathcal{K}$ captures pairwise semantic similarity between all observations across two solutions and is invariant to their ordering. For two solutions $c_i$ and $c_j$, their similarity under kernel $\mathcal{K}$ is

$$\mathcal{K}(c_i, c_j) \propto \sum_{p=1}^{s} \sum_{q=1}^{s} \phi(o_{i,p})^\top \phi(o_{j,q}),$$

where $o_{i,p}$ denotes the $p$-th observation generated from the $p$-th stimulus in the set of $s$ stimuli for $c_i$, and similarly for $o_{j,q}$. $\phi(\cdot)$ denotes the semantic embedding of an observation.

We highlight two implementation details. First, predicting for *all* possible sets of stimuli is not feasible in practice, and a dictionary with 3,000 common English words would already have $3{,}000^s$ sets. To this end, MetaMuse exercises the power-of-two random choices (Mitzenmacher et al., 2001). It randomizes two sets of stimuli from the dictionary, and predicts their expected feedback embeddings. Then, MetaMuse selects the one closest to the target embedding. Second, RSDict-SF is bootstrapped, with $w$ solutions generated by RSDict, in order to start training GPR models.

## 3.3 Developing Stimuli into Executable Solutions

Third, developing executable solutions from $s$ stimuli requires structured **waypoint reasoning**. Waypoints serve as intermediate checkpoints that progressively transform seemingly unrelated stimuli

(§3.2.1) into solutions for the problem at hand. Unlike free-form chain-of-thought (Wei et al., 2023; Mehrotra et al., 2024), we observe that waypoints prevent LLMs from superficially developing solutions, e.g., simply turning stimuli into variable names in code.

MetaMuse currently defines the following waypoints. The first is *property extraction*, where the LLM associates the given stimuli with related concepts and properties. For example, given the keyword "flower," one associated property might be "a ring of petals." These outputs are then passed to the second waypoint, *problem mapping*, where the LLM connects the extracted properties to problem-related observations. Continuing the example, "a ring of petals" may remind the LLM of a circular structure, which can be mapped to a circular buffer in algorithm design. These observations are then fed to the third waypoint, *solution formulation*, where the LLM combines observations and synthesizes the complete description of a new solution. The final waypoint is *code generation*, where the LLM translates the solution description into executable Python code. The waypoint prompts are included in the appendix (§C.2).

# 4 EMPIRICAL RESULTS

**Baselines.** We evaluate MetaMuse against 21 baselines. The LLM-based heuristic design baselines include MCTS-AHD (Zheng et al., 2025), ReEvo (Ye et al., 2024), OpenEvolve (Sharma, 2025), PlanSearch (Wang et al., 2025), and Repeated Sampling (Brown et al., 2024). We also include human-crafted state-of-the-art heuristics for cache replacement, including LRU, LFU, FIFO, Sieve (Zhang et al., 2024), S3FIFO (Yang et al., 2023), TinyLFU (Einziger et al., 2017), SLRU (Huang et al., 2013), Clock (Corbato, 1968), and ARC (Megiddo et al., 2003). For the online bin packing problem, we include Next Fit, Worst Fit, Almost Worst Fit (AWF) (Johnson, 1973), First Fit (Dósa & Sgall, 2013), Best Fit (Dósa & Sgall, 2014), Harmonic-k (Lee & Lee, 1985), and Refined First Fit (RFF) (Yao, 1980). For Harmonic-k, we set $k$=4 to align its number of bin categories with RFF.

**MetaMuse.** We take a dictionary of common English words and remove stop-words to get 2,899 keywords. Each solution is ideated from $s$=4 keywords. The feedback embedding consists of performance measurements on $n$=30 traces. For cache replacement, these $n$ traces are generated by libCacheSim (Yang et al., 2020), from different Zipfian distributions. For bin packing, $n$ traces are from various Weibull distributions. The number of RSDict-SF warmup solutions is $w$=100, roughly one-third of the total solutions in one experiment. Finally, prompts are included in §C.

**Experiment setup.** We focus on two high-impact problems at a global cloud provider: cache replacement and bin packing. In each experiment, all methods aim to ideate and generate 350 executable solutions. To the best of our ability, we configure PlanSearch to output at least 15, 11, and 12 observations, while ideating with GPT-4o, Llama3.3-70B, and DeepSeek-V3, respectively. Many baselines such as MCTS-AHD can tune parameters in their solutions. §5 discusses safeguards against unsafe solutions, and environments are instrumented to catch errors, e.g., long-running execution. Unsafe solutions are re-implemented.

To evaluate cache replacement solutions, we use 96 real-world workload traces from four data access scenarios (Table 1): RetrievalAttention (Liu et al., 2024a) (24 "ra-fwe" and 24 "ra-multikey" traces), Tencent block storage (Zhang et al., 2020; 2018) (24 "tencent-storage" traces), and Alibaba cloud storage (ali, 2022; Li et al., 2020; Wang et al., 2022) (24 "alibaba-storage" traces). The cache capacity is set to 10% of the number of distinct objects in each trace.

To evaluate online bin packing solutions, we use 288 workload traces: BPPLIB library (Delorme et al., 2018) (72 "Falkenauer-U" and 72 "Scholl-1" traces), Weibull distribution with parameter (`shape`=3, `scale`=45) (Castineiras et al., 2012) (72 "Weibull"), and Gaussian distribution with parameter (`mean`=0.3662, `std`=0.1416) (Yan et al., 2022) (72 "Gaussian").

## 4.1 METAMUSE GENERATES HIGH-PERFORMING SOLUTIONS

**Cache replacement.** Figure 3 compares top solutions, as selected by the average cache miss ratio over all 96 workload traces. Box plots show their miss ratio reduction with respect to FIFO.

We first delve into results from GPT-4o. At the $90^{th}$-percentile trace, MetaMuse achieves 5.17%–9.89% lower miss ratio than LLM-based baselines, and 1.75%–13.03% lower than human heuristics. At the $75^{th}$-percentile trace, MetaMuse achieves 3.62%–6.39% lower miss ratio than LLM-based

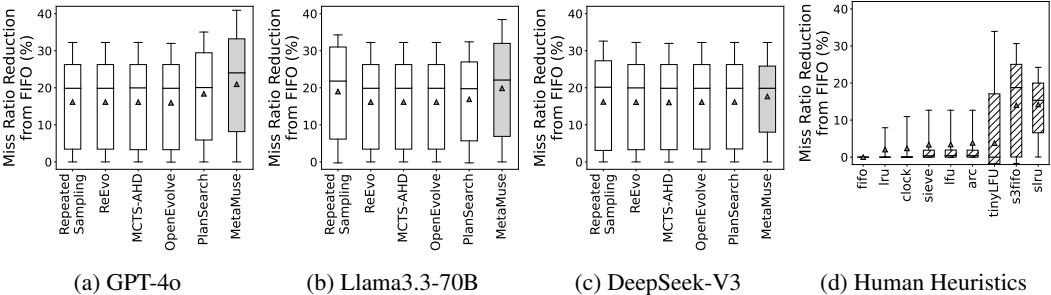

(a) GPT-4o  (b) Llama3.3-70B  (c) DeepSeek-V3  (d) Human Heuristics

Figure 3: Comparison of top cache solutions generated by *MetaMuse* and the baselines. Each box plot represents the best solution from each model and shows the miss ratio reduction (relative to FIFO) across 96 traces. MetaMuse achieves higher reductions across nearly all percentiles and across different LLMs.

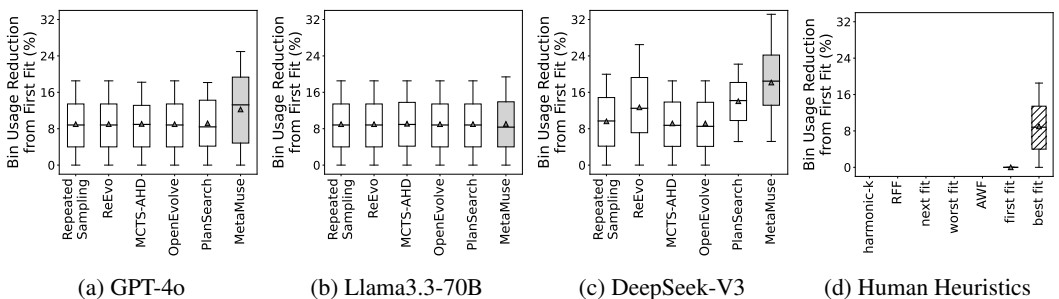

(a) GPT-4o  (b) Llama3.3-70B  (c) DeepSeek-V3  (d) Human Heuristics

Figure 4: Comparison of top online bin packing solutions generated by *MetaMuse* and the baselines. Each box plot represents the best solution from each model and shows the bin usage reduction (relative to First Fit) across 288 traces. MetaMuse achieves higher reductions across nearly all percentiles and across different LLMs. We note that some human-designed heuristics are not visible because they perform worse than First Fit.

baselines, and 6.62%–35.76% lower than human heuristics. We also see improvement with Llama3.3-70B—at the $90^{th}$-percentile trace, MetaMuse achieves 2.41%–7.20% lower miss ratio than LLM-based baselines, and 0.94%–10.34% lower than human heuristics; at the $75^{th}$-percentile trace, MetaMuse achieves 3.67%–5.17% lower miss ratio than LLM-based baselines, and 5.48%–34.62% lower than human heuristics. On DeepSeek-V3, at the 90th-percentile trace, MetaMuse achieves up to 6.05% lower miss ratio than LLM-based baselines, and up to 9.14% lower than human heuristics.

**Online bin packing.** Figure 4 compares top solutions, as selected by the average bin usage over all 96 workload traces. Box plots show their bin usage reduction, with respect to First Fit.

We first delve into results from GPT-4o. At the $90^{th}$-percentile trace, MetaMuse achieves 9.25%–9.42% lower bin usage than LLM-based baselines, and 9.25%–20.59% lower than human heuristics. At the $75^{th}$-percentile trace, MetaMuse achieves 6.94%–7.75% lower bin usage than LLM-based baselines, and 7.56%–18.36% lower than human heuristics. With Llama3-7B, at the $90^{th}$-percentile trace, MetaMuse achieves up to 0.19% less bin usage than LLM-based baselines, and up to 12.50% lower bin usage than human heuristics. With DeepSeek-V3, at the $90^{th}$-percentile trace, MetaMuse achieves up to 21.06% less bin usage than LLM-based baselines, and up to 30.93% lower bin usage than human heuristics.

### 4.2 METAMUSE GENERATES THE MOST DIVERSE SET OF SOLUTIONS

**Cache replacement.** The discovery of useful solutions depends on having diverse solutions. We evaluate diversity by the number of solutions with a distinct feedback embedding. Across all LLMs, MetaMuse consistently achieves higher diversity over LLM-based baselines. From analyzing 350 solutions generated by each method on GPT-4o, MetaMuse has 1.47× more distinct solutions on

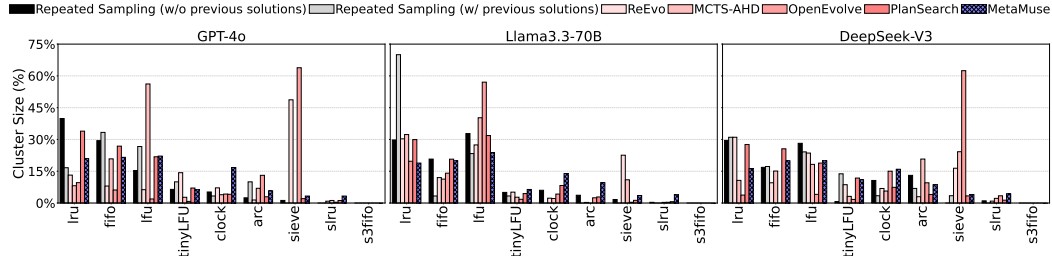

Figure 5: MetaMuse generates a more diverse set of solutions. Using human-designed heuristics as cluster centroids, we find that the cluster sizes of MetaMuse-generated solutions exhibit a lower standard deviation. In contrast, baselines tend to concentrate on a subset of centroids.

average. On DeepSeek-V3, MetaMuse has $1.57\times$ more distinct solutions on average. On Llama3.3-70B, results show that MetaMuse can have $1.78\times$ more distinct solutions on average.

Furthermore, we observe that having a higher diversity translates into lower availability bias. We recall that Repeated Sampling tends to generate solutions that tightly cluster to well-known human heuristics such as LRU, LFU, and FIFO, due to availability bias (§2.3). Using the nine human heuristics as centroids, empirical results show that MetaMuse reduces the cluster density (computed by $\frac{num\_points\_in\_cluster}{max\_euclidean\_distance}$), for nearly all centroids. In other words, MetaMuse solutions are less tightly clustered than Repeated Sampling. For example, MetaMuse reduces the LRU cluster density by 35.28% (i.e., 6.15 down to 3.98). We see cluster density reduction even on the less common heuristics, e.g., the ARC cluster density reduces by 77.45% (i.e., 1.02 down to 0.23).

Finally, Figure 5 uses human heuristics as centroids, and evaluate whether generated solutions tend to cluster to a subset of centroids. Ideally, the cluster sizes should be uniform. The standard deviation of cluster size percentages for MetaMuse is 0.08/0.06/0.06, for GPT-4o/DeepSeek-V3/Llama3-70B. This is much lower than baselines: Repeated Sampling (0.14/0.11/0.11), Repeated Sampling with previous solutions (0.11/0.11/0.11), ReEvo (0.14/0.10/0.10), MCTS-AHD (0.17/0.08/0.08), OpenEvolve (0.19/0.19/0.19), and PlanSearch (0.12/0.10/0.99).

**Online bin packing.** We evaluate diversity by the number of solutions with a distinct feedback embedding. Across all LLMs, MetaMuse consistently achieves a higher diversity over LLM-based baselines. From analyzing 350 solutions generated by each method on GPT-4o, MetaMuse has $1.44\times$ more distinct solutions on average. On DeepSeek-V3, it has $1.80\times$ more distinct solutions on average. On Llama3-70B, it has $1.31\times$ more distinct solutions on average.

## 4.3 METAMUSE INCURS A LOW PER-SOLUTION COST

We view a low per-solution cost as an important factor for driving the adoption of LLMs for creative ideation tasks. In the cache replacement case study, the average cost for MetaMuse to generate *one* full solution is 2.16 cents with GPT-4o, 2.11 cents with DeepSeek-V3, and 2.35 cents with Llama3.3-70B. These costs include using GPT-4o for code generation. For reference, running the Repeated Sampling baseline (with previous solutions) costs an average of 3.38 cents per solution.

Specifically, property extraction and problem mapping consume an average of 589.39/141.54 input/output tokens per solution with GPT-4o, 982.60/233.75 input/output tokens with DeepSeek-V3, and 768.20/207.50 input/output tokens with Llama3.3-70B. Solution formulation consumes an average of 934.04/230.21 input/output tokens per solution with GPT-4o, 473.01/425.77 input/output tokens with DeepSeek-V3, and 409.53/353.10 input/output tokens with Llama3.3-70B. Code generation consumes an average of 1463.92/1190.39 input/output tokens per solution with GPT-4o, 2223.72/1937.29 input/output tokens with DeepSeek-V3, and 2259.62/1979.89 input/output tokens with Llama3.3-70B. We note that if an implementation is considered unsafe (§5), it is re-implemented.

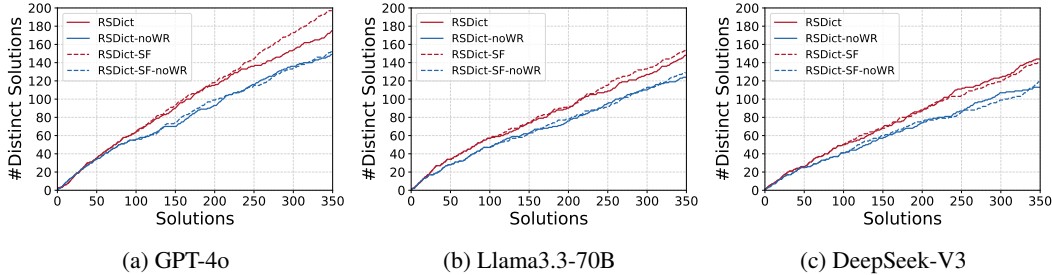

|     |     |     |
| :-: | :-: | :-: |
| (a) GPT-4o | (b) Llama3.3-70B | (c) DeepSeek-V3 |

Figure 6: Comparison of the diversity of cache replacement solutions generated by different stimuli selection strategies: RSDict and RSDict-SF. "-noWR" denotes the removal of waypoint reasoning. Ideation is performed with different LLMs, while code generation is handled by GPT-4o. RSDict-SF (dashed lines) outperforms RSDict (solid lines). Removing waypoint reasoning (blue lines) reduces solution diversity compared with the full approaches (red lines).

## 4.4 Surprising Designs from MetaMuse

In addition to quantitative evaluations, we highlight MetaMuse-generated designs that are not immediately obvious to engineers. To do so, we look at top-performing cache replacement solution: *MetaMuse-533* (§D.1) and *MetaMuse-488* (§D.2).

One is the counter named NSE in MetaMuse-533. NSE deviates from the common belief of favoring newly admitted objects. Instead, it tracks the number of times that a cached object has witnessed eviction events. It is used in the score calculation, along with recent access time and frequency counter. Effectively, MetaMuse-533 behaves to favor objects that have remained longer in the cache (up to some thresholds). Our engineers hypothesize that NSE helps to learn objects' access patterns and prevent thrashing (Denning, 1968).

Another is the use of hashing functions to segment the cache space in MetaMuse-488, which differs from the more common multi-tiered architecture. MetaMuse-488 tries to maintain different groups of cache objects (as identified by their key hashes), and exercises per-group replacement policies in different segments. In contrast, engineers tend to use multi-tiered architecture, as a way to differentiate the importance of cache objects.

Finally, we highlight the use of saturating counters in MetaMuse-533. They increment until reaching a pre-defined threshold. MetaMuse-533 deviates from conventional usage scenarios, which are to reduce storage overhead by capping the maximum value of a counter (Yang et al., 2023; Corbato, 1968). Instead, MetaMuse-533 seems to use saturating counters to accumulate meaningful usage history, but not so much as to mislead eviction decisions. For example, a saturating frequency counter allows MetaMuse-533 to identify bursty objects, while preventing their counts from growing too large during the burst.

## 4.5 Ablation Study

We test key components in MetaMuse using the case study of cache replacement algorithms.

**Stimuli selection strategies.** We test RSDict and RSDict-SF by running experiments on GPT-4o. First, the majority of RSDict and RSDict-SF solutions outperform Repeated Sampling solutions, and this demonstrates the value of having stimuli for steering ideation. Comparing solutions for the $90^{th}$-percentile workload trace, 67.20% of RSDict solutions and 75.60% of RSDict-SF solutions have a higher cache hit ratio than Repeated Sampling. In addition, these hit ratios can be up to 13.13% and 27.07% higher, respectively. Similarly, for the $75^{th}$-percentile workload trace, we see 66.80% of RSDict solutions and 72.40% of RSDict-SF solutions outperform Repeated Sampling. In addition, these hit ratios can be up to 15.43% and 21.93% higher, respectively.

Second, compared with RSDict, RSDict-SF has a greater number of solutions that can outperform Repeated Sampling. Empirical results show 12.50% and 8.38% more solutions, with respect to the $90^{th}$ percentile and $75^{th}$ percentile of workload traces, respectively. Furthermore, looking at solution diversity, RSDict-SF results in 13.17% more distinct solutions than RSDict.

**Waypoint reasoning.** Figure 6 shows that waypoint reasoning enables MetaMuse to generate more diverse solutions. We fix the coding agents to GPT-4o. At each iteration, we instrument MetaMuse to generate solutions with and without waypoint reasoning. When ideating with GPT-4o (Figure 6a), waypoint reasoning improves the number of distinct solutions from 149 to 175 for RSDict, and from 152 to 197 for RSDict-SF. When ideating with DeepSeek-V3 (Figure 6c), waypoint reasoning improves the number of distinct solutions from 113 to 144 for RSDict, and from 119 to 140 for RSDict-SF. When ideating with Llama3.3-70B (Figure 6b), waypoint reasoning improves the number of distinct solutions from 124 to 148 for RSDict, and from 129 to 154 for RSDict-SF.

## 5 DISCUSSION

**Safeguards against unsafe solutions.** In the case of system algorithms, three main errors can result in unsafe implementations: *(1)* long-running executions (e.g., exceeding 5 seconds), *(2)* excessive memory usage beyond user-specified limits, and *(3)* illegal behaviors (e.g., falsely claiming a cache hit when the requested object is absent). Our benchmarking environments monitor the per-run execution time and peak memory usage when evaluating each solution. Both can be collected from the GNU `time` command. To detect illegal behaviors, we validate the returned object value against the expected value in the trace. From our experience, $\sim$2.28% of solutions generated by MetaMuse are classified as unsafe and subsequently discarded.

Finally, with Python scripts as the output, it is possible to leverage libraries in the Python ecosystem. We discuss two ongoing efforts. One is performing unit tests, with libraries such as unittest (uni, 2025). Another is the design-code equivalence; we are exploring the use of LLMs to generate state machines from design descriptions and code implementations for the equivalence check.

**Limitations and future discussion.** First, our implementation explores an instantiation of external stimuli (§3.2), which is specifically chosen for its generality and problem-agnostic nature. However, the MetaMuse framework is extensible to take in other instantiations, and future work will explore possibilities such as long-term task-related memories. Second, while our work currently focuses on fully automated ideation, an interesting direction is to explore whether subjective human guidance (e.g., human expertise) could further improve the solution usefulness. Third, although we evaluate on real-world workload traces (§4), deploying LLM-generated algorithms in business-critical production environments remains an ongoing collaboration with engineers at a global cloud provider. We anticipate that lessons from this deployment effort will inform deeper practical impact.

**Algorithm benchmarking costs.** The RSDict-SF strategy considers the feedback embeddings of previous solutions (§3.2.1), which requires benchmarking each solution on $n$ workload traces. Such benchmarking can take a significant amount of time and resources, especially if they involve running an end-to-end system with long workload traces. If costs are prohibitively high, RSDict-SF may impact the practicality of MetaMuse. We note, however, that reducing benchmarking overhead is an active area of research in the systems community Akram & Sawalha (2019), with techniques such as discrete-event simulations, hardware acceleration, and parallelization. At the same time, we are exploring stimuli selection strategies that reduce reliance on extensive benchmarking.

**Synergy with existing approaches.** Although MetaMuse emphasizes creative leaps in the solution space, we recognize the value in related approaches that iteratively refine algorithm designs (Liu et al., 2024b; Ye et al., 2024; Zheng et al., 2025; Romera-Paredes et al., 2024; Novikov et al., 2025; Sharma, 2025). The two paradigms are complementary, in a way similar to human's divergent and convergent thinking processes. MetaMuse can provide the initial population of solution candidates, which can then be further improved by the iterative refinement approach.

## 6 CONCLUSION

To practically drive algorithm generation, MetaMuse is a framework of creative ideation that mitigates availability bias in LLMs. MetaMuse combines three self-reflection principles, guiding the stages of creative ideation. Evaluations show that it can generate high-performing solutions for two high-impact problems at a global cloud provider: cache replacement and bin packing algorithms. Furthermore, we observe surprising design considerations not immediately obvious to human engineers.

**Reproducibility statement.** Key prompts are included in the appendix (§C). The code and datasets are publicly available at `https://github.com/illinois-nsai/MetaMuse`.

**Ethics statement.** This work poses no ethical issues.

**Acknowledgments.** We thank Encheng Xie (University of Illinois Urbana-Champaign) for his valuable discussions and contributions to the experiments.

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

## A    LLM USAGE STATEMENT

LLMs did not play a significant role in the research ideation or writing process (beyond grammar checking) and are therefore not considered contributors.

## B    CACHE WORKLOAD TRACES

We detail the traces for evaluation on cache replacement in Table 1 (§4). To show their access patterns, we plot one trace from each data access scenario (Figure 7). The $x$-axis denotes a virtual timestamp, which is the number of cache accesses thus far. The $y$-axis denotes the accessed objects. Their IDs are sequentially assigned, following the chronological order of the time they appear for the first time. The intervals between the gray vertical lines equal the cache capacity, which is set as 10% of the trace footprint (i.e., the number of distinct object IDs).

| scenario | ra-fwe | ra-multikey | tencent-storage | alibaba-storage |
|---|---|---|---|---|
| release year | 2024 | 2024 | 2020 | 2024 |
| avg #cache accesses | 10598.42 | 27390.71 | 53503.79 | 7582.08 |
| avg #distinct objects | 938.54 | 1092.29 | 225.125 | 4108.83 |

Table 1: The data access scenarios of cache replacement used in this work. Each scenario consists of 24 cache traces.

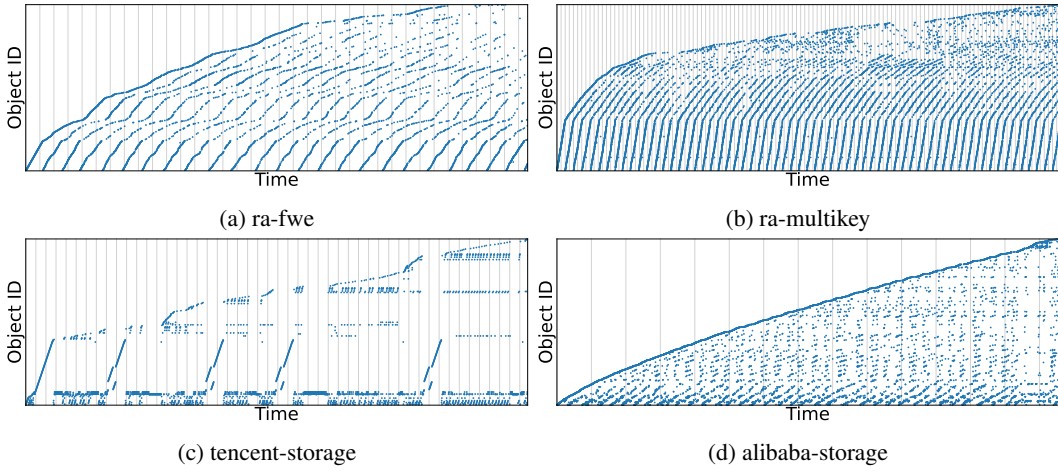

(a) ra-fwe

(b) ra-multikey

(c) tencent-storage

(d) alibaba-storage

Figure 7: The access pattern of one cache trace from each data access scenario.

## C    PROMPTS

### C.1    IDEATION PROMPTS

Repeated Sampling uses the following prompt for solution formulation, generating its natural language description:

---

**The Ideation Prompt for Repeated Sampling**

You are an expert in computer systems. Your task is to create an innovative cache replacement policy.
Provide your creative policy using the following JSON structure:
{
    "metadata": "use a few sentences to summarize the metadata specifically
              maintained by the policy here",

---

```
    "evict": "use a few sentences to summarize how the policy chooses the
             eviction victim here",
    "update_after_hit": "use a few sentences to summarize how the policy
                         updates **each** of the metadata it maintains
                         immediately after a cache hit here",
    "update_after_insert": "use a few sentences to summarize how the policy
                            updates **each** of the metadata it maintains
                            immediately after inserting a new object into
                            the cache here",
    "update_after_evict": "use a few sentences to summarize how the policy
                           updates **each** of the metadata it maintains
                           immediately after evicting the victim here"
}
```

Do not include any additional text or explanation in your response.

We use PlanSearch's original prompts (Wang et al., 2025) to generate observations. These observations are then inserted into the placeholder `[[hints]]` of the following prompt for solution formulation:

---

**The Ideation Prompt for PlanSearch**

You are an expert in computer systems. Create an innovative cache replacement policy using the following hints:

[[hints]]

Provide your creative policy using the following JSON structure:

```
{
    "metadata": "use a few sentences to summarize the metadata specifically
                 maintained by the policy here",
    "evict": "use a few sentences to summarize how the policy chooses the
              eviction victim here",
    "update_after_hit": "use a few sentences to summarize how the policy
                         updates **each** of the metadata it maintains
                         immediately after a cache hit here",
    "update_after_insert": "use a few sentences to summarize how the policy
                            updates **each** of the metadata it maintains
                            immediately after inserting a new object into the
                            cache here",
    "update_after_evict": "use a few sentences to summarize how the policy
                           updates **each** of the metadata it maintains
                           immediately after evicting the victim here"
}
```

Do not include any additional text or explanation in your response.

---

MetaMuse uses the following prompt for property extraction and problem mapping. They are the first two waypoints (§3.3).

---

**The Waypoint Reasoning Prompt for MetaMuse**

You are an expert in computer systems. Your task is to use the concepts inspired by the given word or phrase to design creative ideas for cache replacement policies.
Answer in the following format:
<The given word or phrase> relates to the concept of <concept 1>.
<Concept 1> relates to <Concept 2>.
...
<Concept n-1> relates to <Concept n>
Inspired by <Concept n>, <your creative ideas for cache replacement policies in a few sentences>.
**Example 1**:
The given word or phrase: angular momentum.
Answer:
"Angular momentum" relates to the concept of "rotation".
"Rotation" relates to "cycle".
Inspired by "cycle", a cyclic pointer can be maintained to track cached objects and determine eviction victims.

---

---

**Example 2**:
The given word or phrase: zebra.
Answer:
"Zebra" relates to the concept of "stripe".
"Stripe" relates to "segmentation".
Inspired by "segmentation", a cache can be divided into segments with different eviction priorities, and each segment can use a distinct eviction policy.
**The given word or phrase**: [[word]]
Do not include any additional text or explanation in your answer.

---

For solution formulation, MetaMuse uses the same ideation prompt as PlanSearch. The placeholder `[[hints]]` will be replaced with the observations derived from the first two waypoints.

## C.2 CODING PROMPTS

All methods use the following prompt to implement a solution in Python. The placeholders, `[[metadata]]`, `[[evict]]`, `[[update_after_hit]]`, `[[update_after_insert]]`, and `[[update_after_evict]]`, will be filled in with the corresponding fields in the JSON-formatted output of the ideation prompts. `[[design]]` will be filled with a concatenation of these fields.

---

**The Coding Prompt**

You are an expert in Python. Your task is to implement a deterministic cache replacement policy in Python without any randomness. You can only reference the attributes provided below. You have read-only access to these attributes and no access to any functions.

Accessible attributes:
An "object" represents the unit of a request, such as inserting an object into the cache or retrieving an object from the cache. Each object 'obj' provides the following **read-only** attributes that you can reference:
- 'obj.key' (str): A string that uniquely identifies the object.
- 'obj.size' (int): A positive integer representing the size of the object in bytes.
You can also reference the following **read-only** attributes provided by a cache snapshots 'cache_snapshot' :
- 'cache_snapshot.cache' (dict): A dictionary containing the cached objects, where the keys are the objects'keys, and the values are the corresponding objects themselves.
- 'cache_snapshot.size' (int): A non-negative integer representing the current total size of the cache in bytes.
- 'cache_snapshot.capacity' (int): A positive integer representing the maximum allowed size of the cache in bytes.
- 'cache_snapshot.access_count' (int): The current total number of cache accesses. You can also use this to represent current time.
- 'cache_snapshot.hit_count' (int): The current total number of cache hits.
- 'cache_snapshot.miss_count' (int): The current total number of cache misses.
The cache replacement policy you need to implement is described below:

Cache replacement policy: [[design]]

Implement this policy using the Python code framework below. Your implementation must strictly follow the comments in this Python code framework.

Code framework: ...
You **must not** alter the provided code framework. Also, keep the two comments "# Put tunable constant parameters below" and "# Put the metadata specifically maintained by the policy below" unchanged. Wrap your code with "'python and "' and include nothing else in your answer.

---

The code framework is shown below:

```python
# Import anything you need below. You must not use any randomness.
# For example, you cannot `import random`. Also, you cannot use any
# function in `numpy` that uses randomness, such as the functions in
# `numpy.random`. Put tunable constant parameters below
# Put the metadata specifically maintained by the policy below. [[metadata]]
```

```python
def evict(cache_snapshot, obj):
    '''
    This function defines how the policy chooses the eviction victim.
    [[evict]]
    - Args:
        - `cache_snapshot`: A snapshot of the current cache state.
        - `obj`: The new object that needs to be inserted into the cache.
    - Return:
        - `candid_obj_key`: The key of the cached object that will be evicted
          to make room for `obj`.
    '''
    candid_obj_key = None
    # Your code below

    return candid_obj_key

def update_after_hit(cache_snapshot, obj):
    '''
    This function defines how the policy update the metadata it maintains
    immediately after a cache hit.
    [[update_after_hit]]
    - Args:
        - `cache_snapshot`: A snapshot of the current cache state.
        - `obj`: The object accessed during the cache hit.
    - Return: `None`
    '''
    # Your code below

def update_after_insert(cache_snapshot, obj):
    '''
    This function defines how the policy updates the metadata it maintains
    immediately after inserting a new object into the cache.
    [[update_after_insert]]
    - Args:
        - `cache_snapshot`: A snapshot of the current cache state.
        - `obj`: The object that was just inserted into the cache.
    - Return: `None`
    '''
    # Your code below

def update_after_evict(cache_snapshot, obj, evicted_obj):
    '''
    This function defines how the policy updates the metadata it maintains
    immediately after evicting the victim.
    [[update_after_evict]]
    - Args:
        - `cache_snapshot`: A snapshot of the current cache state.
        - `obj`: The object to be inserted into the cache.
        - `evicted_obj`: The object that was just evicted from the cache.
    - Return: `None`
    '''
    # Your code below
```

# D  CASE STUDIES

## D.1  METAMUSE-533

```python
# Import anything you need below. You must not use any randomness.
# For example, you cannot `import random`. Also, you cannot use any function
# in `numpy` that uses randomness, such as the functions in `numpy.random`.

# Put tunable constant parameters below
```

```python
MAX_PREDICTIVE_SCORE = 100
MAX_ENTROPY = 100
MAX_NEURAL_ALIGNMENT = 100
MAX_ACCESS_FREQUENCY = 100
MAX_RECENCY = 100
MAX_DIFFERENTIAL_PRIVACY_NOISE = 100

# Put the metadata specifically maintained by the policy below.
# The policy maintains predictive likelihood scores, stochastic model outputs,
# data entropy values, neural alignment scores, access frequency, recency,
# differential privacy noise factors, quantum error correction codes,
# and deep reinforcement learning model state-action values.
metadata = {
    'predictive_likelihood_scores': {},
    'stochastic_model_outputs': {},
    'data_entropy_values': {},
    'neural_alignment_scores': {},
    'access_frequency': {},
    'recency': {},
    'differential_privacy_noise_factors': {},
    'quantum_error_correction_codes': {},
    'deep_rl_state_action_values': {}
}

def evict(cache_snapshot, obj):
    '''
    This function defines how the policy chooses the eviction victim.
    The policy chooses the eviction victim by combining predictive likelihood
    scores, stochastic model outputs, data entropy values, neural alignment
    scores, access frequency, recency, and differential privacy noise factors,
    adjusted by the deep reinforcement learning model's recommendations to
    balance performance, privacy, and future access predictions.
    - Args:
        - `cache_snapshot`: A snapshot of the current cache state.
        - `obj`: The new object that needs to be inserted into the cache.
    - Return:
        - `candid_obj_key`: The key of the cached object that will be evicted
            to make room for `obj`.
    '''
    candid_obj_key = None
    min_score = float('inf')

    for key, cached_obj in cache_snapshot.cache.items():
        score = (
            metadata['predictive_likelihood_scores'].get(key, 0) +
            metadata['stochastic_model_outputs'].get(key, 0) +
            metadata['data_entropy_values'].get(key, 0) +
            metadata['neural_alignment_scores'].get(key, 0) +
            metadata['access_frequency'].get(key, 0) +
            metadata['recency'].get(key, 0) +
            metadata['differential_privacy_noise_factors'].get(key, 0)
        )

        if score < min_score:
            min_score = score
            candid_obj_key = key

    return candid_obj_key

def update_after_hit(cache_snapshot, obj):
    '''
    This function defines how the policy updates the metadata it maintains
    immediately after a cache hit. After a cache hit, the policy increases
    the predictive likelihood score, updates the stochastic model, recalculates
    data entropy, adjusts the neural alignment score, updates access frequency
```

```
    and recency, recalculates the differential privacy noise factor, and updates
    the state-action values in the deep reinforcement learning model.
    - Args:
        - `cache_snapshot`: A snapshot of the current cache state.
        - `obj`: The object accessed during the cache hit.
    - Return: `None`
    '''
    key = obj.key
    metadata['predictive_likelihood_scores'][key] = min(
        metadata['predictive_likelihood_scores'].get(key, 0) + 1,
        MAX_PREDICTIVE_SCORE)
    metadata['stochastic_model_outputs'][key] = min(
        metadata['stochastic_model_outputs'].get(key, 0) + 1,
        MAX_PREDICTIVE_SCORE)
    metadata['data_entropy_values'][key] = min(
        metadata['data_entropy_values'].get(key, 0) + 1,
        MAX_ENTROPY)
    metadata['neural_alignment_scores'][key] = min(
        metadata['neural_alignment_scores'].get(key, 0) + 1,
        MAX_NEURAL_ALIGNMENT)
    metadata['access_frequency'][key] = min(
        metadata['access_frequency'].get(key, 0) + 1, MAX_ACCESS_FREQUENCY)
    metadata['recency'][key] = cache_snapshot.access_count
    metadata['differential_privacy_noise_factors'][key] = min(
        metadata['differential_privacy_noise_factors'].get(key, 0) + 1,
        MAX_DIFFERENTIAL_PRIVACY_NOISE)
    metadata['deep_rl_state_action_values'][key] = min(
        metadata['deep_rl_state_action_values'].get(key, 0) + 1,
        MAX_PREDICTIVE_SCORE)

def update_after_insert(cache_snapshot, obj):
    '''
    This function defines how the policy updates the metadata it maintains
    immediately after inserting a new object into the cache.
    After inserting a new object, the policy initializes the predictive
    likelihood score, updates the stochastic model, calculates initial data
    entropy, sets the neural alignment score, initializes access frequency and
    recency, assigns a differential privacy noise factor, generates quantum
    error correction codes, and updates the deep reinforcement learning model
    to include the new state.
    - Args:
        - `cache_snapshot`: A snapshot of the current cache state.
        - `obj`: The object that was just inserted into the cache.
    - Return: `None`
    '''
    key = obj.key
    metadata['predictive_likelihood_scores'][key] = 1
    metadata['stochastic_model_outputs'][key] = 1
    metadata['data_entropy_values'][key] = 1
    metadata['neural_alignment_scores'][key] = 1
    metadata['access_frequency'][key] = 1
    metadata['recency'][key] = cache_snapshot.access_count
    metadata['differential_privacy_noise_factors'][key] = 1
    metadata['quantum_error_correction_codes'][key] = 1
    metadata['deep_rl_state_action_values'][key] = 1

def update_after_evict(cache_snapshot, obj, evicted_obj):
    '''
    This function defines how the policy updates the metadata it maintains
    immediately after evicting the victim.
    After evicting a victim, the policy removes its metadata, updates the
    stochastic model, recalculates data entropy for remaining entries, adjusts
    neural alignment scores, adjusts differential privacy noise factors, updates
    quantum error correction codes, and retrains the deep reinforcement learning
    model to adapt to the new cache state.
```

```
    - Args:
        - `cache_snapshot`: A snapshot of the current cache state.
        - `obj`: The object to be inserted into the cache.
        - `evicted_obj`: The object that was just evicted from the cache.
    - Return: `None`
    '''
    evicted_key = evicted_obj.key
    if evicted_key in metadata['predictive_likelihood_scores']:
        del metadata['predictive_likelihood_scores'][evicted_key]
    if evicted_key in metadata['stochastic_model_outputs']:
        del metadata['stochastic_model_outputs'][evicted_key]
    if evicted_key in metadata['data_entropy_values']:
        del metadata['data_entropy_values'][evicted_key]
    if evicted_key in metadata['neural_alignment_scores']:
        del metadata['neural_alignment_scores'][evicted_key]
    if evicted_key in metadata['access_frequency']:
        del metadata['access_frequency'][evicted_key]
    if evicted_key in metadata['recency']:
        del metadata['recency'][evicted_key]
    if evicted_key in metadata['differential_privacy_noise_factors']:
        del metadata['differential_privacy_noise_factors'][evicted_key]
    if evicted_key in metadata['quantum_error_correction_codes']:
        del metadata['quantum_error_correction_codes'][evicted_key]
    if evicted_key in metadata['deep_rl_state_action_values']:
        del metadata['deep_rl_state_action_values'][evicted_key]

    # Recalculate data entropy for remaining entries
    for key in cache_snapshot.cache:
        metadata['data_entropy_values'][key] = min(
            metadata['data_entropy_values'].get(key, 0) + 1,
            MAX_ENTROPY)

    # Adjust neural alignment scores
    for key in cache_snapshot.cache:
        metadata['neural_alignment_scores'][key] = min(
            metadata['neural_alignment_scores'].get(key, 0) + 1,
            MAX_NEURAL_ALIGNMENT)

    # Adjust differential privacy noise factors
    for key in cache_snapshot.cache:
        metadata['differential_privacy_noise_factors'][key] = min(
            metadata['differential_privacy_noise_factors'].get(key, 0) + 1,
            MAX_DIFFERENTIAL_PRIVACY_NOISE)

    # Update quantum error correction codes
    for key in cache_snapshot.cache:
        metadata['quantum_error_correction_codes'][key] = min(
            metadata['quantum_error_correction_codes'].get(key, 0) + 1,
            MAX_PREDICTIVE_SCORE)

    # Retrain the deep reinforcement learning model
    for key in cache_snapshot.cache:
        metadata['deep_rl_state_action_values'][key] = min(
            metadata['deep_rl_state_action_values'].get(key, 0) + 1,
            MAX_PREDICTIVE_SCORE)
```

## D.2   METAMUSE-488

```
PARTITION_COUNT = 3  # Number of partitions based on usage patterns
INITIAL_PRIORITY = 1
INITIAL_RARITY = 1

metadata = {
```

```
        'partitions': [{} for _ in range(PARTITION_COUNT)],  # List of dictionaries for
        each partition
        'priority_scores': {},  # Dictionary mapping obj.key to priority score
        'rarity_scores': {},  # Dictionary mapping obj.key to rarity score
        'timestamps': {},  # Dictionary mapping obj.key to access timestamps
}

def evict(cache_snapshot, obj):
    candid_obj_key = None
    for partition in metadata['partitions']:
        # Find the least priority and rarity score item in the partition
        candidates = sorted(
            partition.items(),
            key=lambda item: (
                metadata['priority_scores'][item[0]],
                metadata['rarity_scores'][item[0]],
                metadata['timestamps'][item[0]]
            )
        )
        if candidates:
            candid_obj_key = candidates[0][0]
            break

    return candid_obj_key

def update_after_hit(cache_snapshot, obj):
    metadata['timestamps'][obj.key] = cache_snapshot.access_count
    metadata['rarity_scores'][obj.key] += 1  # Increase rarity score based on access
     frequency
    # Reevaluate priority within the partition
    # Here we assume a simple priority adjustment based on access frequency
    metadata['priority_scores'][obj.key] = max(
        metadata['priority_scores'][obj.key], metadata['rarity_scores'][obj.key])

def update_after_insert(cache_snapshot, obj):
    partition_index = hash(obj.key) % PARTITION_COUNT
    metadata['partitions'][partition_index][obj.key] = obj
    metadata['priority_scores'][obj.key] = INITIAL_PRIORITY
    metadata['rarity_scores'][obj.key] = INITIAL_RARITY
    metadata['timestamps'][obj.key] = cache_snapshot.access_count

def update_after_evict(cache_snapshot, obj, evicted_obj):
    partition_index = hash(evicted_obj.key) % PARTITION_COUNT
    if evicted_obj.key in metadata['partitions'][partition_index]:
        del metadata['partitions'][partition_index][evicted_obj.key]
        del metadata['priority_scores'][evicted_obj.key]
        del metadata['rarity_scores'][evicted_obj.key]
        del metadata['timestamps'][evicted_obj.key]

    # Rebalance partitions based on current usage patterns
    for partition in metadata['partitions']:
        for key in partition:
            # Simplistic strategy to adjust priority and rarity scores
            metadata['priority_scores'][key] = max(
                INITIAL_PRIORITY, metadata['priority_scores'][key] - 1)
            metadata['rarity_scores'][key] = max(
                INITIAL_RARITY, metadata['rarity_scores'][key] - 1)
```

## D.3    COMPARE RSDICT AND RSDICT-NOWR USING AN EXAMPLE

We compare RSDict and RSDict-noWR with a concrete example on cache replacement to illustrate waypoint reasoning.

RSDict and RSDict-noWR both use the stimuli set {best, impose, extra, pale} to generate cache replacement policies.

### D.3.1 PROPERTY EXTRACTION

RSDict reasons about the problem-related property of each stimulus:

- best → comparison → ranking
- impose → authority → hierarchy
- extra → surplus → overflow
- pale → fading → decay

### D.3.2 PROBLEM MAPPING

RSDict maps the extracted properties to the problem domain to generate observations:

- ranking → objects in the cache can be ranked based on access frequency or recency, with the lowest-ranked objects being evicted first
- hierarchy → a hierarchical cache replacement policy can be developed where objects are ranked based on importance, and lower-ranked objects are evicted first
- overflow → a cache replacement policy can focus on evicting the least valuable entries to make room for critical ones that exceed the cache's immediate capacity, essentially creating a priority-based overflow management system
- decay → a cache replacement policy can use a decay function to gradually decrease the priority of cache entries, selecting the least 'bright' or least relevant items for eviction

### D.3.3 SOLUTION FORMULATION

RSDict designs a solution using the observations:

- metadata: The policy maintains metadata including access frequency, recency of access, hierarchical importance, and a decay score for each cache entry. A dual ranking system assigns a composite score based on these factors.
- update_after_hit: The policy calculates a composite score using access frequency, recency, importance rank, and decay values to identify the lowest scored entries for eviction, prioritizing those near a threshold in the hierarchical cache structure.
- update_after_insert: After a cache hit, the access frequency and recency values are incremented, the decay function is reset or slowed, and hierarchical importance may be updated to reflect improved relevance.
- update_after_evict: Post eviction, the metadata recalibrates the hierarchical importance thresholds, updates decay dynamics, and modifies frequency/recency metrics for remaining entries to ensure balanced prioritization.

RSDict-noWR directly uses the given stimuli to generate a solution:

- metadata: The policy maintains metadata including a "Pale Access Frequency" score for each entry, representing its likelihood of future accesses, and an "Extra Usage Counter" to track rare intense usage patterns. Additionally, it includes 'Best Recent Utilization' timestamps, recording the time since the last access.
- evict: The policy chooses the eviction victim by identifying the entry with the lowest "Pale Access Frequency" score, the lowest 'Extra Usage Counter', and the oldest 'Best Recent Utilization' timestamp, focusing on entries least likely to be used in the near future.
- update_after_hit: Upon a cache hit, the policy increases the 'Pale Access Frequency' score and "Extra Usage Counter" for the accessed entry, boosting its likelihood to stay, and updates the 'Best Recent Utilization' timestamp to the current time.
- update_after_insert: After inserting a new object, the policy initializes its "Pale Access Frequency" score and "Extra Usage Counter" to default low values, indicating an unbiased entry yet to prove its importance, and sets the 'Best Recent Utilization' timestamp to the current time.

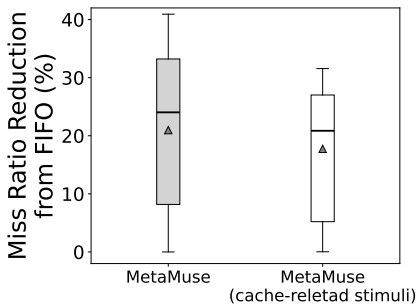

Figure 8: Comparison of top cache solutions generated by *MetaMuse* using stimuli from an English dictionary and *MetaMuse* using cache-related stimuli. Each box plot represents the best solution from each model and shows the miss ratio reduction (relative to FIFO) across 96 traces. Using stimuli from an English dictionary achieves higher reductions across nearly all percentiles.

- update_after_evict: Upon eviction, the policy resets the metadata for the evicted entry, ensuring that new insertions don't inherit old, irrelevant frequencies or utilization timestamps, and recalibrates other entries to maintain consistent scoring dynamics.

This example shows that without waypoint reasoning, LLMs tend to use the given stimuli in a superficial way. In RSDict-noWR, LLMs uses the stimulus "pale" to name a metadata "Pale Access Frequency", which is identical to a standard frequency counter. In RSDict, LLMs first derive the concept of "decay" from "pale", then maps it to observation "decay function", which is more difficult to come up with compared with frequency counting in cache replacement.

### D.3.4 DIVERSITY EVALUATION

After calculating the feedback embeddings of the two solutions, we find that RSDict-noWR's solution is identical to LFU, while RSDict's solution is different from any human heuristic.

## E IMPACTS OF USING TASK-RELATED STIMULI

§3.2 mentions that stimuli can be unbiased to the problem, forcing LLMs to associate with knowledge that seems probabilistically irrelevant. And, one domain-agnostic instantiation of stimuli is keywords from an English dictionary. However, MetaMuse can also accommodate task-related stimuli. To this end, this section presents empirical results on the impacts of using task-related stimuli.

Our experiments are based on designing cache replacement policies with GPT-4o. We start by prompting GPT-4o to output a list of cache-related stimuli, and GPT-4o generates 78 keywords: *lfru*, *consistency*, *ghost*, *clock*, *cache*, *optimal*, *associativity*, *s3-fifo*, *benchmark*, *bélády*, *policy*, *hit*, *temporal*, *memory*, *slru*, *queues*, *queue*, *arc*, *static*, *replacement*, *coherence*, *algorithms*, *access*, *plru*, *miss*, *brrip*, *probabilistic*, *caching*, *storage*, *sieve*, *prefetching*, *tree*, *lirs*, *analysis*, *timestamp*, *lifo*, *latency*, *policies*, *dynamic*, *recency*, *algorithm*, *reinsertion*, *performance*, *tlru*, *flash*, *clock-pro*, *ratio*, *discard*, *binary*, *fifo*, *filo*, *expiration*, *mq*, *hawkeye*, *distance*, *srrip*, *pollution*, *ssds*, *locality*, *lfuda*, *eviction*, *reuse*, *drrip*, *mru*, *lfu*, *prediction*, *pointers*, *lru*, *inter-reference*, *rrip*, *approximation*, *distributed*, *throughput*, *data*, *aging*, *survival*, *metadata*, *streaming*. Then, we feed these stimuli to MetaMuse, and generate 350 cache replacement algorithm designs. Empirical results show that task-related stimuli reduce the number of distinct solutions by 11.

Figure 8 compares top solutions, as selected by the average cache miss ratio over all 96 workload traces. Box plots show their miss ratio reduction, with respect to FIFO heuristics.

At the $90^{th}$-percentile trace, MetaMuse using stimuli from an English dictionary achieves 3.24% lower miss ratio than MetaMuse using cache-related stimuli. At the $75^{th}$-percentile trace, MetaMuse using stimuli from an English dictionary achieves 6.99% lower miss ratio than MetaMuse using cache-related stimuli.

