# OpenReview forum: "MetaMuse: Algorithm Generation via Creative Ideation"
_ICLR.cc/2026/Conference — ICLR 2026 Poster_

### Official Review · Reviewer_eCqA · 2025-10-29

**Soundness:** 3
**Presentation:** 3
**Contribution:** 3
**Rating:** 6
**Confidence:** 4

**Summary:**

The goal is to generate algorithms with LLMs. The paper notes that diversity is necessary for generating good algorithms, but that LLMs are inherently bad at creating diverse output. A method is thus proposed to encourage LLMs to generate diverse outputs using a rather clever mechanism that adds a completely unrelated set of words to the prompt input. The method uses a Gaussian Process Regression (GPR) model that tries to find a set of words that will result in diverse outputs. The method is tested on a "cache replacement" problem and an online bin packing problem.

**Strengths:**

1. The idea is simple and effective.
2. The paper follows a strong line of argumentation that goes through the entire work.
3. The paper is relatively easy to understand (with one small exception, see below) and I think the method should be fairly easy to implement for anyone else.

**Weaknesses:**

The main weaknesses of this paper lie in the experiments.

1. The problems chosen are kind of strange picks, since many of the approaches compared against do not use these two problems.
2. The training is apparently cheap, but only two problems are compared against, which does not give me so much confidence in the method.
3. The waypoint reasoning is not entirely clear to me. Maybe I am overthinking it, but I do not entirely understand the flow of the prompts leading to an algorithm. The flowchart in Figure 2 is not helping -- certainly you do not just prompt with "flower". Please not that any text in the Appendix about this is insufficient -- this needs to be clear in the main paper.
4. The results for the online bin packing are somewhat unsatisfying, as I do not see any state-of-the-art methods discussed or compared against. Beating a bunch of simple heuristics does not seem like the best argument for this work.
5. I also wonder why only online algorithms are chosen here rather than the problems in, e.g,. ReEvo or OpenEvolve.

**Questions:**

1. See point about waypoints.
2. See last point of weaknesses.

---

> ### Author Response · Authors · 2025-11-23
> **Response to Reviewer eCqA**
>
> > The problems chosen are kind of strange picks, since many of the approaches compared against do not use these two problems.
>
> > The training is apparently cheap, but only two problems are compared against, which does not give me so much confidence in the method.
>
> Thank you for the two comments above. Our project goal specifically targets problems that are of high-value and high-impact to cloud business. We believe that solving real-world problems would demonstrate LLM’s potential, to the systems community and industry. Considering the problems in the paper, bin-packing impacts the server utilization, and caching is in all levels of the cloud stack to improve data access.
>
> Furthermore, we would like to point out that MetaMuse’s workflow is not designed for any particular system algorithm problem. From our experience, switching to a new problem (e.g., caching problem to bin-packing problem) does not require redesigning MetaMuse’s workflow. Given its generality, we are working with engineers to identify more high-value and high-impact system algorithm problems faced in production.
>
> > The waypoint reasoning is not entirely clear to me. Maybe I am overthinking it, but I do not entirely understand the flow of the prompts leading to an algorithm. The flowchart in Figure 2 is not helping -- certainly you do not just prompt with "flower". Please note that any text in the Appendix about this is insufficient -- this needs to be clear in the main paper.
>
> Thank you for this comment. You are right that we do not just prompt with keywords such as “flower”. In addition to including our prompts in Appendix C, we have rewritten Section 3.3. For example, Appendix C shows that the prompt includes both the task descriptions and the keywords.
>
> > The results for the online bin packing are somewhat unsatisfying, as I do not see any state-of-the-art methods discussed or compared against. Beating a bunch of simple heuristics does not seem like the best argument for this work.
>
> Thank you for this comment. Our baselines include solutions commonly deployed by real- world cloud providers, in addition to SOTA solutions that we know of. These baselines are also used in the publications of related system algorithm design efforts. Furthermore, we also include LLM-based heuristics design baselines.
>
> > I also wonder why only online algorithms are chosen here rather than the problems in, e.g., ReEvo or OpenEvolve.
>
> Thank you for this comment. Many real-world problems fall under online algorithms, especially that it is really difficult to accurately estimate future user workloads and requests. Therefore, many scenarios adapt online algorithms. Online bin-packing problem is an example, as it is difficult to know what resources (e.g., virtual machines) that users will request in the future. Furthermore, the complexity of online problems can be beyond human reasoning, as such our project goal is to see whether LLMs can design better algorithms for these real-world problems.

---

> > ### Comment · Reviewer_eCqA · 2025-11-25
> >
> > I have a major problem here - if you want to target online algorithms specifically, that's completely fine, but it ought to be right in the title of the paper. Your title implies a general method, but that's not what you seem to want to provide. Fine -- but change the title, abstract and introduction. Make it 100% clear that this approach is only targeting online algorithms.
> >
> > Regarding waypoints in 3.3, the rewritten text is unfortunately just as unclear as before. The examples in the appendix actually make the reasoning quite clear, I am not sure why you can't just describe the prompt or provide some text that actually describes what is happening. 3.3 is just so overcomplicated.
> >
> > Regarding the heuristics: I have investigated the heuristics compared against in more detail and feel comfortable with them being sota.
> >
> > Can the authors also comment on the quality / complexity of the code that was generated versus the human-designed heuristics?

---

> > > ### Author Response · Authors · 2025-11-28
> > > **Response to Reviewer eCqA**
> > >
> > > > I have a major problem here - if you want to target online algorithms specifically, that's completely fine, but it ought to be right in the title of the paper. Your title implies a general method, but that's not what you seem to want to provide. Fine -- but change the title, abstract and introduction. Make it 100% clear that this approach is only targeting online algorithms.
> > >
> > > Thank you for this comment. We have changed the abstract, and contributions in the introduction.
> > >
> > > > Regarding waypoints in 3.3, the rewritten text is unfortunately just as unclear as before. The examples in the appendix actually make the reasoning quite clear, I am not sure why you can't just describe the prompt or provide some text that actually describes what is happening. 3.3 is just so overcomplicated.
> > >
> > > Thank you for this comment. Section 3.3 provides a running example of “*flower*” in between texts, to illustrate the different waypoints. In addition, we have tried moving prompts from Appendix C to the main text, but they are too long. So, we decided to leave prompts in Appendix C.
> > >
> > > > Can the authors also comment on the quality / complexity of the code that was generated versus the human-designed heuristics?
> > >
> > > Thank you for this comment. First, we evaluate quality by the algorithm performance. Code quality can also mean metrics such as readability, line of codes, and so on. However, these metrics are not the scope of our paper. Second, we evaluate complexity by the runtime time and space costs, as described in Section 5’s “*Safeguards against unsafe solutions*”.

---

> > > > ### Comment · Reviewer_eCqA · 2025-11-28
> > > >
> > > > I think the title is still much too broad, although I do not know if that can be changed at this stage.
> > > >
> > > > Regarding 3.3, I still do not think it is clear enough for anyone who does not already know about the topic. It is simply unclear from this text whether there are multiple prompts or a single prompt, and how it is prompted.
> > > >
> > > > Regarding code quality, I am not sure how that is "out of scope." Just because you don't want to do it does not make it out of scope -- out of scope requires an argument as to why it is irrelevant or not necessary.
> > > >
> > > > To summarize: I think the idea in this paper is very interesting. I am just not sure if I can move my score up. I will continue to consider it.

---

> > > > > ### Author Response · Authors · 2025-11-28
> > > > >
> > > > > > Regarding 3.3, I still do not think it is clear enough for anyone who does not already know about the topic. It is simply unclear from this text whether there are multiple prompts or a single prompt, and how it is prompted.
> > > > >
> > > > > Thank you. We would like to see how to better address this comment. Each waypoint has its own prompt, with the previous prompt's output as part of the prompt. Is this what you are looking for?
> > > > >
> > > > > > Regarding code quality, I am not sure how that is "out of scope." Just because you don't want to do it does not make it out of scope -- out of scope requires an argument as to why it is irrelevant or not necessary.
> > > > >
> > > > > Thank you. We would like to clarify "out of scope". Code quality is a broad term, and we focus on the metric of performance. Performance is what end-users care the most, as it is what they can actually "feel" (and even "pay for" in some business service-level agreements). The other metrics (e.g., readability and lines of code) are not as commonly requested by end-users, in our experience. For example, end-users would not care if a cache algorithm were implemented in 1,000 lines or 1,00 lines, as long as it runs fine and achieves good performance.
> > > > >
> > > > > We hope that this explanation is helpful, and we are happy to answer follow-up questions.

---

### Official Review · Reviewer_7Wx3 · 2025-10-31

**Soundness:** 4
**Presentation:** 3
**Contribution:** 4
**Rating:** 8
**Confidence:** 3

**Summary:**

MetaMuse offers a general framework for helping LLMs break free of their inherent availability bias, in order to generate meaningfully diverse, creative solutions to difficult problems -- particularly those with complex, non-linear, discontinuous landscapes within and/or between the design and performance spaces. MetaMuse describes three principles of self-reflection that can help diversify LLM responses by activating and applying knowledge from probabilistically unrelated concepts ("external stimuli"). The principles (1) explicitly measure existing solution diversity, and (2) use these metrics to select external stimuli that steer future iterations toward different approaches. Given a set of stimuli, MetaMuse (3) develops an executable implementation using waypoint reasoning, which encourages non-trivial analysis and application of the external stimuli in the context of the problem at hand. The authors demonstrate the efficacy and generalizability of their approach by showing strong performance across several different models and two real-world problem domains (cache replacement and bin packing).

**Strengths:**

The paper is well written, and the approach within is well motivated. The proposed principles of ideation and self-reflection are thought provoking, and serve as a great baseline for systematizing the elusive concept of creative ideation. The ideas are intentionally described in a general manner, and could easily be applied or adapted to many exciting domains. The present manuscript already offers two compelling case studies, with extensive baseline comparisons that highlight MetaMuse's strong performance on both. The ablation studies offer strong evidence that the proposed principles are effective and essential for achieving the stated performance. Overall, this paper offers considerable value and impact for the scientific community.

**Weaknesses:**

1. The authors partially motivate their work by citing that previous solutions ignore higher-level design parameters like data structure selection, in favor of tweaks to the scoring function. Yet, the authors measure diversity in performance/feedback space, and explicitly dismiss the efficacy of observing semantic differences in language or code, including data structure changes (l. 200-205). It seems like the emphasis on performance-only diversity metrics would also fail to incentivize/meaningfully reward the higher-level tweaks, but these larger jumps in algorithmic design space feel critical. How often do you witness higher-level changes in your experiments? How much of the increased diversity stems from scoring function changes (or other tweaks mirroring previous works' exhibited tendencies) vs. more novel types of changes? Would you consider the examples discussed in Section 4.4 to be scoring function changes, or larger-scope ones (this might be clear to a system algorithms expert, but wasn't clear to me)? I would be interested to see more discussion on this point.

2. The coding safeguards are referenced several times (e.g. l. 299, 393), but the relevant discussion in Section 5 is quite limited; I like the idea quite a lot, but I expected more based on the earlier mentions. To what extent are these safeguards actually realized vs. hypothetical? How often are they used/required in your experiments? I would like to see this piece either clarified (with a detailed description, perhaps in the appendix) or tempered in the earlier references.

**Questions:**

Please see weaknesses. Additionally:

1. Figure 1 / l. 137 - is the clustering algorithm such that it forms a partition of the 1200 solutions, based on the closest human-derived heuristic in the set? i.e., if something is sufficiently unique (like the solutions described in l.399-409), would it have to belong to the nearest cluster, or would it be separately labeled? If it's the former, the mere presence of the cluster isn't necessarily compelling -- it's there by construction. We'd have to know what the radius/desnity pattern of the cluster was, similar to the discussion you present in l. 376. Perhaps a visualization like a scatter plot might be more illustrative in that case. Relatedly, as someone with little knowledge about cache replacement, I'm curious whether the bias toward LRU/LFU/FIFO might reflect something deeper than mere availability/prevalence bias. For example: since you're measuring by similarity of hit ratios, do the algorithms with large clusters have some inherent performance advantage? This might explain the demonstrated bias in a few ways: maybe it makes sense for the LLM to offer variants of these, since they perform well; alternatively, maybe the metric causes highly performant algorithms cluster near these baselines, even if the underlying approach is different.

2. l. 210 -- how important is it that the performance metrics be bounded? What would be required to accommodate unbounded metrics? Relatedly, how well could this approach generalize to multiple metrics? What are the scaling considerations?

3. l. 259 -- when using RSDict-RF, are the $w$ RSDict solutions for bootstrapping done as a preprocessing step before the experiment (such that $c_0$ is based on RSDict-RF stimuli) or are they used as part of the exploration (such that $c_0$ ... $c_w$ are the result of RSDict and only $c_{w+1}$ stem from RSDict-RF stimuli)?

4. Figures 3/4 -- could you clarify the phrase "top...solutions generated by..."? Specifically, is each box plot representative of the single best solution from each model (where the displayed variance comes from that single solution performing over 96/288 traces), or does each box plot reflect the top n solutions for each model (each performing over the 96/288 traces, then also averaged together)?


## Minor comments
- l. 87 - the antecedent of "This" is unclear
- l. 137 - "centroids of human heuristics" was unclear to me until reading the explanation in l. 375.
- l. 244 - "in embedding" --> "in the (feedback) embedding"
- l. 399 - NSE is not defined; it would be helpful to expand the acronym or provide a short explanation for context

---

> ### Author Response · Authors · 2025-11-23
> **Response to Reviewer 7Wx3**
>
> > The authors partially motivate their work by citing that previous solutions ignore higher-level design parameters like data structure selection, in favor of tweaks to the scoring function. Yet, the authors measure diversity in performance/feedback space, and explicitly dismiss the efficacy of observing semantic differences in language or code, including data structure changes (l. 200-205). It seems like the emphasis on performance-only diversity metrics would also fail to incentivize/meaningfully reward the higher-level tweaks, but these larger jumps in algorithmic design space feel critical.
>
> Thank you for this comment. You are right that we measure diversity in the performance feedback space. After all, this is what end users care about. However, the deeper question is how do you create a set of diverse solutions that would hopefully cover the space? In the context of MetaMuse, the performance feedback isn’t really used to incentivize, rather its purpose is to guide subsequent ideations to explore other regions of the space. In addition to performance feedback, we also tried to use semantic embedding to differentiate semantic differences in language or code. However, as mentioned in Section 3.1, this approach does not work well.
>
> > Would you consider the examples discussed in Section 4.4 to be scoring function changes, or larger-scope ones (this might be clear to a system algorithms expert, but wasn't clear to me)? I would be interested to see more discussion on this point.
>
> Thank you for this comment. Examples discussed in Section 4.4 go beyond scoring function changes. Specifically, the saturating counter relates to designing the behavior of a counter, and the NSE counter highlights a different dimension of metadata to track. On the other hand, scoring function changes commonly refer to changing the weights and mathematical operators in the score calculation of cache objects. Furthermore, we added another surprising design to Section 4.4. Specifically, *MetaMuse-488* is designed to be segmented with hash functions, to assign cache objects to different segments.
>
> > The coding safeguards are referenced several times (e.g. l. 299, 393), but the relevant discussion in Section 5 is quite limited; I like the idea quite a lot, but I expected more based on the earlier mentions. To what extent are these safeguards actually realized vs. hypothetical? How often are they used/required in your experiments? I would like to see this piece either clarified (with a detailed description, perhaps in the appendix) or tempered in the earlier references.
>
> Thank you for this comment. We have rewritten the “*Safeguards against unsafe solutions*” paragraph in Discussion. We have realized checks for the three three main errors: (1) long-running execution, (2) excessive memory usage over the user-specified requirement, and (3) illegal behaviors. Furthermore, we also describe on-going efforts, e.g., Python unit tests, and design-and-code equivalence check.

---

> > ### Author Response · Authors · 2025-11-23
> > **Response to Reviewer 7Wx3 (Continued)**
> >
> > > Figure 1 / l. 137 - is the clustering algorithm such that it forms a partition of the 1200 solutions, based on the closest human-derived heuristic in the set? i.e., if something is sufficiently unique (like the solutions described in l.399-409), would it have to belong to the nearest cluster, or would it be separately labeled? If it's the former, the mere presence of the cluster isn't necessarily compelling -- it's there by construction. We'd have to know what the radius/density pattern of the cluster was, similar to the discussion you present in l. 376. Perhaps a visualization like a scatter plot might be more illustrative in that case.
> >
> > Thank you for this comment. For Figure 1, the clustering algorithm considers the distance to the closest human heuristics. Ideally, the distribution of cluster sizes should be uniform, which would suggest the set of generated solutions are diverse. To evaluate this uniformness, we performed additional experiments on MetaMuse and baselines. Then, Section 4.2 calculates the standard deviation of cluster size percentages. Results show that MetaMuse has a lower standard deviation than LLM-based baselines, which suggest that it has a more uniform distribution.
> >
> > > Relatedly, as someone with little knowledge about cache replacement, I'm curious whether the bias toward LRU/LFU/FIFO might reflect something deeper than mere availability/prevalence bias. For example: since you're measuring by similarity of hit ratios, do the algorithms with large clusters have some inherent performance advantage? This might explain the demonstrated bias in a few ways: maybe it makes sense for the LLM to offer variants of these, since they perform well; alternatively, maybe the metric causes highly performant algorithms cluster near these baselines, even if the underlying approach is different.
> >
> > Thank you for this comment. This is an interesting thought. We would like to point out that Figure 1 is based on 30 traces, which are randomly generated. In other words, the set of traces should not favor certain human heuristics. Furthermore, the goal of MetaMuse is to generate diverse solutions. Keeping multiple solutions (or variants) that perform similarly would mean that we keep redundant solutions. Also, since end users care about performance only, it should not matter which one we keep.
> >
> > > l. 210 -- how important is it that the performance metrics be bounded? What would be required to accommodate unbounded metrics? Relatedly, how well could this approach generalize to multiple metrics? What are the scaling considerations?
> >
> > Thank you for this comment. These are very interesting questions. Boundness is not really the reason that our approach works, but it prevents the approach from breaking in practice. If the performance metrics are not bounded, then MetaMuse would be dealing with an infinite space of performance feedback embedding. It is not clear how this infinite space can be efficiently represented. The problem exacerbates, as we consider that there can be many performance metrics in the feedback embedding.
> >
> > Multiple metrics should work fine, as MetaMuse independently evaluates each dimension of the feedback embedding. This brings some scaling considerations. Particularly, the number of GPR models grows with the number of dimensions in the feedback embedding. As a result, the training/prediction costs grow as well.
> >
> > > l. 259 -- when using RSDict-RF, are the RSDict solutions for bootstrapping done as a preprocessing step before the experiment (such that is based on RSDict-RF stimuli) or are they used as part of the exploration (such that are the result of RSDict and only stem from RSDict-RF stimuli)?
> >
> > Thank you for this comment. Solutions generated in the bootstrapping step are also counted as part of RSDict-SF. We have updated the texts in Section 3.2.1 accordingly.
> >
> > > Figures 3/4 -- could you clarify the phrase "top...solutions generated by..."? Specifically, is each box plot representative of the single best solution from each model (where the displayed variance comes from that single solution performing over 96/288 traces), or does each box plot reflect the top n solutions for each model (each performing over the 96/288 traces, then also averaged together)?
> >
> > Thank you for this comment. It is the single best solution from each model. We have updated Figure 3 and 4 captions accordingly

---

### Official Review · Reviewer_wtEQ · 2025-11-01

**Soundness:** 3
**Presentation:** 3
**Contribution:** 2
**Rating:** 4
**Confidence:** 4

**Summary:**

The paper investigates the challenge of algorithm generation using Large Language Models (LLMs), specifically addressing the issue where LLMs tend to generate known heuristics due to "availability bias." The authors propose a framework named MetaMuse to overcome this bias by evaluating diversity in the performance space, steering ideation with external stimuli, and employing structured waypoint reasoning. Experiments demonstrate that this approach outperforms LLM baselines and human-designed algorithms on cache replacement and online bin packing problems.

**Strengths:**

Automatic algorithm design is a frontier in LLM applications, and the paper's exploration of LLM's creative limitations is highly valuable.
The proposed "external stimuli" and "waypoint reasoning" offer a novel approach to enhance LLM performance on algorithmic tasks, distinct from traditional evolutionary methods.

**Weaknesses:**

1. The paper fails to specify the exact version numbers or model checkpoints for the public APIs used (e.g., GPT-40). Given the rapid iteration of LLM models, this makes the experimental results nearly impossible to reproduce accurately.

2. While the paper provides a final monetary cost estimate in §4.3, this analysis is superficial. It completely overlooks the analysis of token consumption, a core driver of cost. This omission prevents readers from assessing the framework's true computational overhead and token efficiency, and precludes any meaningful cost-benefit comparison with other methods.

3. The effectiveness of external stimuli lacks theoretical support, and its success might be coincidental. Its generalizability and stability are questionable, making it seem more like a sophisticated random search than a reliable method for creative ideation.

4. The comparison with state-of-the-art baselines like OpenEvolve is methodologically shallow. The paper's central thesis is that MetaMuse overcomes availability bias while existing methods do not. However, the comparison only presents final performance metrics without providing evidence to support this core claim. A convincing comparison should have analyzed and shown whether the solutions from baselines like OpenEvolve indeed cluster around known heuristics as theorized, while MetaMuse successfully explores a broader, more novel solution space.

**Questions:**

1. Code Correctness Verification: The generated code's safety (e.g., against timeouts) is mentioned. How was the logical correctness of the code (i.e., does the implementation correctly match the algorithm's description) verified? What was the first-pass success rate of the code generation step?

2. Experimental Cost Analysis: Could you provide a detailed breakdown of the token costs for generating a single solution with MetaMuse, including prompt and completion tokens for each stage? How does the framework's token efficiency compare to a simple repeated sampling baseline?

3. API Versioning: To ensure reproducibility, please provide the specific API version numbers or model release dates for GPT-40, Llama3.3-70B, and DeepSeek-V3 used in the experiments.

---

> ### Author Response · Authors · 2025-11-23
> **Response to Reviewer wtEQ**
>
> > Code Correctness Verification: The generated code's safety (e.g., against timeouts) is mentioned. How was the logical correctness of the code (i.e., does the implementation correctly match the algorithm's description) verified? What was the first-pass success rate of the code generation step?
>
> Thank you for this comment. This is a very interesting question. We have realized checks for the three three main errors: (1) long-running execution, (2) excessive memory usage over the user-specified requirement, and (3) illegal behaviors. The failure rate of the code generation step is about 2.28% Furthermore, we have an on-going effort on leveraging LLM to check the design-and-code equivalence. We have rewritten “*Safeguards against unsafe solutions*” in Discussion.
>
> > The paper fails to specify the exact version numbers or model checkpoints for the public APIs used (e.g., GPT-40). Given the rapid iteration of LLM models, this makes the experimental results nearly impossible to reproduce accurately.
>
> > API Versioning: To ensure reproducibility, please provide the specific API version numbers or model release dates for GPT-4o, Llama3.3-70B, and DeepSeek-V3 used in the experiments.
>
> Thank you for the two comments above. We have updated the texts in Section 2.3, to include the LLM model versions. In addition, we will include the API version when we publish the paper, due to potential anonymity concerns.
>
> > While the paper provides a final monetary cost estimate in §4.3, this analysis is superficial. It completely overlooks the analysis of token consumption, a core driver of cost. This omission prevents readers from assessing the framework's true computational overhead and token efficiency, and precludes any meaningful cost-benefit comparison with other methods.
>
> > Experimental Cost Analysis: Could you provide a detailed breakdown of the token costs for generating a single solution with MetaMuse, including prompt and completion tokens for each stage? How does the framework's token efficiency compare to a simple repeated sampling baseline?
>
> Thank you for the two comments above. We have updated the texts in Section 4.3, to include the number of input/output tokens, for different stages of the MetaMuse workflow, for different LLM models. For comparisons, we have also included the monetary cost estimate for repeated sampling baseline (with previous solutions in the prompt).
>
> > The comparison with state-of-the-art baselines like OpenEvolve is methodologically shallow. The paper's central thesis is that MetaMuse overcomes availability bias while existing methods do not. However, the comparison only presents final performance metrics without providing evidence to support this core claim. A convincing comparison should have analyzed and shown whether the solutions from baselines like OpenEvolve indeed cluster around known heuristics as theorized, while MetaMuse successfully explores a broader, more novel solution space.
>
> Thank you for this comment. We have performed additional experiments and added Figure 5, to show that MetaMuse has a more uniform distribution than LLM-based baselines. Particularly, we calculate the standard deviation of cluster size percentages, for MetaMuse and baselines. MetaMuse has a lower standard deviation than LLM-based baselines, which suggest that it has a more uniform distribution.
>
> > The effectiveness of external stimuli lacks theoretical support, and its success might be coincidental. Its generalizability and stability are questionable, making it seem more like a sophisticated random search than a reliable method for creative ideation.
>
> Thank you for this comment. This is an interesting question on the theoretical aspect. The process of developing external stimuli into executable solutions rely on LLM’s internal capability (e.g., reasoning and association) and pre-trained knowledge. As such, it is hard to provide theoretical support, to guarantee that two different stimuli would definitely result in two different solutions. This is also the reason that MetaMuse targets creative ideation, which is a process of generating a stream of ideas (rather than one single solution). In other words, looking at a stream of generated ideas, the goal is for MetaMuse to maximize the chance of having diverse ideas. Having said that, we welcome public ideas on the theoretical aspect.

---

### Official Review · Reviewer_kD1D · 2025-11-02

**Soundness:** 2
**Presentation:** 3
**Contribution:** 3
**Rating:** 4
**Confidence:** 4

**Summary:**

This paper proposes MetaMuse to address the lack of creativity in LLMs when applied to algorithm design. The framework controls the LLM to generate novel algorithms through the use of external "keywords" and a structured chain-of-thought (Waypoint Reasoning). The method demonstrates notable performance improvements on the two evaluated system problems: cache replacement and online bin packing.

**Strengths:**

1. The paper is well-structured and easy to understand. The authors exhibit a deep and accurate understanding of the diversity problem inherent in LLM-based algorithm design.

2. The use of performance as a feature for algorithms is a smart and feasible approach. This overcomes the difficulty in clustering and analysis caused by the discontinuous nature of the algorithm space.

3. The attribute extraction and problem mapping of the keywords are interesting and make sense. The experiments successfully demonstrate that this external stimulation mechanism significantly increases the diversity of algorithms generated by the LLM.

4. The paper is largely successful in its argument and introduces a degree of novelty to the field of LLM-driven algorithm design.

**Weaknesses:**

1. Missing Mechanism for Incremental Improvement: The framework overly emphasizes "Eureka" ideas to improve algorithm quality. However, iterative algorithm design often requires continuity and gradual improvement. Significant gains often come from small adjustments (e.g., hyperparameter tuning) to a good algorithm. MetaMuse currently lacks a mechanism to effectively capture and execute this type of small-scale, continuous refinement, focusing only on large, creative leaps.

2. Insufficient Justification for "Unbiased Stimuli": The authors state that "Stimuli should be unbiased to the problem, forcing LLMs to associate with knowledge that seems probabilistically irrelevant." While this is experimentally successful, the theoretical justification needs strengthening. Why are completely neutral keywords more effective than carefully curated, domain-relevant concepts in avoiding the availability bias? As demonstrated in the authors' examples, unrelated keywords must still first be linked to domain-relevant concepts before further reasoning can occur. Furthermore, the authors should supplement the experiments by showing the performance of non-neutral keywords to demonstrate the necessity of using unbiased stimuli, and discuss the risk of generating incoherent algorithms due to unrelated stimuli.

3. Scalability Concerns with RSDict-SF: The RSDict-SF strategy relies on building a performance prediction model (GPR), which requires a large amount of multi-dimensional feedback embedding data. The practicality of RSDict-SF is limited for domain problems where each evaluation is prohibitively expensive. The paper should offer a more in-depth discussion on how MetaMuse can be adapted to handle sparse feedback or difficult-to-quantify tasks.

4.  Limited Application Scope: The framework is only validated on two specific system problems (cache replacement and online bin packing). Its robustness and generalizability to other types of algorithmic design problems remain to be thoroughly verified.

5. Missing Hyperparameter Analysis: The paper lacks analysis regarding the determination and sensitivity of the framework's hyperparameters. This omission impacts the reproducibility and reliability of the reported results.

**Questions:**

1. Please thoroughly check the paper for adherence to formatting guidelines. For example, acronyms like LRU should only be defined upon their first occurrence.

2. FunSearch and EoH have shown promising results in the Online Bin Packing problem. Why were these LLM-based approaches not included in the experimental comparison? This absence makes it difficult to assess MetaMuse's actual advantage over current advanced techniques.

3. In Figure 1, please clarify how Repeated sampling with previous solutions is specifically implemented. Does the input of previous solutions include all prior sampling results, or only a recent subset? Furthermore, a detailed comparison with this baseline is more crucial than a simple Repeated Sampling analysis to substantiate the paper's claim regarding diversity.

4. Figure 1 illustrates the failure of LLMs to explore the algorithm space effectively. Please clarify what an ideal sampling distribution looks like under the authors' claims (I assume a uniform distribution across different clusters?). More importantly, the authors should display the actual sampling points of MetaMuse to demonstrate that the framework successfully overcomes local clustering and achieves effective exploration of the discontinuous space.

5. The diversity evaluation in MetaMuse is strongly correlated with the distribution and number of traces used to generate the performance feedback embeddings, which directly impacts the clustering results. What is the effect of these factors on MetaMuse? The authors should consider and present how the performance of the generated algorithms is affected by the variance and quantity of the evaluation traces.

---

> ### Author Response · Authors · 2025-11-23
> **Response to Reviewer kD1D**
>
> > Missing Mechanism for Incremental Improvement: The framework overly emphasizes "Eureka" ideas to improve algorithm quality. However, iterative algorithm design often requires continuity and gradual improvement. Significant gains often come from small adjustments (e.g., hyperparameter tuning) to a good algorithm. MetaMuse currently lacks a mechanism to effectively capture and execute this type of small-scale, continuous refinement, focusing only on large, creative leaps.
>
> Thank you for this comment. You are right that this paper’s scope is on “Eureka” ideas. However, we do recognize the value in related approaches that iteratively refine algorithm designs. In fact, both approaches are orthogonal, in a way similar to human’s divergent and convergent thinking processes. MetaMuse can provide the initial population of solution candidates, which can then be further improved by the iterative refinement approach.
>
> We have updated Section 5, to add a paragraph called “*Synergy with existing approaches*”.
>
> > Insufficient Justification for "Unbiased Stimuli": The authors state that "Stimuli should be unbiased to the problem, forcing LLMs to associate with knowledge that seems probabilistically irrelevant." While this is experimentally successful, the theoretical justification needs strengthening. Why are completely neutral keywords more effective than carefully curated, domain-relevant concepts in avoiding the availability bias? As demonstrated in the authors' examples, unrelated keywords must still first be linked to domain-relevant concepts before further reasoning can occur. Furthermore, the authors should supplement the experiments by showing the performance of non-neutral keywords to demonstrate the necessity of using unbiased stimuli, and discuss the risk of generating incoherent algorithms due to unrelated stimuli.
>
> Thank you for this comment. The intuition is that carefully-curated and domain-relevant keywords are basically derived from previously known solutions. However, we agree that it is possible to carefully curate keywords from little-known (or even rarely-known) solutions, which might induce LLMs to ideate less well-known solutions. In other words, you would need human experts or LLMs to try to curate these keywords, for each task or problem.
>
> We have toned down Section 3.2. Instead of saying “Stimuli should be unbiased to the problem”, the text now says “Stimuli can be unbiased to the problem”. In addition, we have added new experiments in Appendix E, to evaluate the impacts of using task-related (i.e., domain-relevant) stimuli. Empirical results show that using task-related stimuli results in 11 less distinct solutions, out of 350 generated solutions.
>
> > Scalability Concerns with RSDict-SF: The RSDict-SF strategy relies on building a performance prediction model (GPR), which requires a large amount of multi-dimensional feedback embedding data. The practicality of RSDict-SF is limited for domain problems where each evaluation is prohibitively expensive. The paper should offer a more in-depth discussion on how MetaMuse can be adapted to handle sparse feedback or difficult-to-quantify tasks.
>
> Thank you for this comment. Since RSDict-SF considers previous solutions in the feedback embedding, it requires benchmarking solutions on *n* workload traces. As you mentioned, benchmarks can take a significant amount of time and resources, especially if they involve running an end-to-end system with long workload traces. If costs are prohibitively high, RSDict-SF may impact the practicality of MetaMuse. We note that benchmarking cost is a problem that the systems community has been actively addressing, and there are techniques ranging from discrete-event simulations, hardware acceleration, to parallelization. At the same time, we are exploring stimuli selection strategies that have a lower requirement on running benchmarks.
>
> We have updated Section 5, to add a paragraph called “*Algorithm benchmarking costs*”.

---

> > ### Author Response · Authors · 2025-11-23
> > **Response to Reviewer kD1D (Continued)**
> >
> > > Limited Application Scope: The framework is only validated on two specific system problems (cache replacement and online bin packing). Its robustness and generalizability to other types of algorithmic design problems remain to be thoroughly verified.
> >
> > Thank you for this comment. Our project goal specifically targets problems that are of high-value and high-impact to cloud business. We believe that solving real-world problems would demonstrate LLM’s potential, to the systems community and industry. Considering the problems in the paper, bin-packing impacts the server utilization, and caching is in all levels of the cloud stack to improve data access. Furthermore, we would like to point out that MetaMuse’s workflow is not designed for any particular system algorithm problem. From our experience, switching to a new problem (e.g., caching problem to bin-packing problem) does not require redesigning MetaMuse’s workflow. Given its generality, we are working with engineers to identify more high-value and high-impact system algorithm problems faced in production.
> >
> > > Please thoroughly check the paper for adherence to formatting guidelines. For example, acronyms like LRU should only be defined upon their first occurrence.
> >
> > Thank you for this comment. We have fixed the acronyms problem you pointed out.
> >
> > > FunSearch and EoH have shown promising results in the Online Bin Packing problem. Why were these LLM-based approaches not included in the experimental comparison? This absence makes it difficult to assess MetaMuse's actual advantage over current advanced techniques.
> >
> > Thank you for this comment. Both FunSearch and AlphaEvolve are from Google Deepmind, but AlphaEvolve (2025) is an extension of FunSearch (2023). Therefore, our evaluation uses the open-sourced version of AlphaEvolve, which is called OpenEvolve. EoH has been shown to be outperformed by more recent efforts, and please refer to MCTS-AHD Table 3 (https://arxiv.org/pdf/2501.08603).
> >
> > > In Figure 1, please clarify how Repeated sampling with previous solutions is specifically implemented. Does the input of previous solutions include all prior sampling results, or only a recent subset?
> >
> > Thank you for this comment. Repeated sampling with previous solutions includes the descriptions and codes of all prior sampling results. We have updated the texts in Section 2.3 accordingly.
> >
> > > Furthermore, a detailed comparison with this baseline is more crucial than a simple Repeated Sampling analysis to substantiate the paper's claim regarding diversity.
> >
> > > Figure 1 illustrates the failure of LLMs to explore the algorithm space effectively. Please clarify what an ideal sampling distribution looks like under the authors' claims (I assume a uniform distribution across different clusters?). More importantly, the authors should display the actual sampling points of MetaMuse to demonstrate that the framework successfully overcomes local clustering and achieves effective exploration of the discontinuous space.
> >
> > Thank you for the two comments above. The ideal distribution of cluster sizes should be close to being uniform. We have added this clarification to Section 2.3.
> >
> > We have performed additional experiments and added Figure 5, to show that MetaMuse has much more uniform distribution than baselines. In addition, we calculate the standard deviation of cluster size percentages, for MetaMuse and baselines. MetaMuse has a lower standard deviation than LLM-based baselines, which suggest that it indeed has a more uniform distribution.
> >
> > > The diversity evaluation in MetaMuse is strongly correlated with the distribution and number of traces used to generate the performance feedback embeddings, which directly impacts the clustering results. What is the effect of these factors on MetaMuse? The authors should consider and present how the performance of the generated algorithms is affected by the variance and quantity of the evaluation traces.
> >
> > Thank you for this comment. You are right that we are using performance feedback embedding to evaluate diversity. This is the reason that we want the traces to be different, (i.e., the performance feedback embedding would cover a wide range of behavior). Otherwise, as you mentioned, the diversity evaluation would be correlated with the bias in the performance feedback embedding. One way is to randomly synthesize traces with tools from the systems community (e.g., libCacheSim for cache workloads). We have updated the texts in Section 3.1 accordingly.

---

### Author Response · Authors · 2025-12-03

We thank reviewers and ACs, for reviewing our submission. Although the discussion period was cut short, we have addressed reviewers’ comments. For your reference, we summarize the major discussion points and paper improvements.

**Availability Bias.** First, we have added clarifications in the text, to address reviewer questions regarding the ideal distribution (```kD1D```), repeated sampling baselines (```kD1D```), and bias towards LRU/LFU/FIFO (```7Wx3```). Second, we have conducted new experiments, to address reviewer requests to see the bias of all baselines, as illustrated in the new Figure 5 (```kD1D```, ```wtEQ```). In addition, to better help reviewers to understand Figure 5, we have rewritten Section 4.2 and added numbers such as the standard deviation of cluster size percentages.

**Safeguards.** For completeness, our submission’s Discussion section talks about safeguards, which detect unsafe implementations. Following reviewer comments, we have added the requested error rates (```wtEQ```), and clarified the deployed and in-progress safeguards (```wtEQ```, ```7Wx3```)

**Scenarios of System Algorithms.** Reviewers (```eCqA```, ```kD1D```) pointed out that the paper presents results from designing two system algorithms: caching and bin-packing. We have explained in our responses — From our experience at a global cloud provider, we specifically target problems that are of high-value and high-impact to the industry. For example, bin-packing impacts the server utilization, and caching is in all levels of the infrastructure to improve data latency. We believe that solving real-world problems would demonstrate LLM’s potential, to the systems community and industry. Having said that, we would like to point out that MetaMuse’s workflow is not designed for any particular system algorithm problem. Given its generality, we are working with engineers to target more high-value and high-impact system algorithm problems in production.

**Costs.** Reviewers raised points regarding two types of costs. The first is the algorithm benchmark costs (```kD1D```), as the RFDict-SF strategy requires benchmarking previous solutions. For this, we have added the “*Algorithm benchmarking costs*” paragraph to the Discussion section. The second is LLM API consumption. Following reviewer’s suggestion (```wtEQ```), we have added the number of input/output tokens consumed in Section 4.3.

**Unbiased Stimuli.** Reviewers raised the point on whether problem-irrelevant keywords are absolutely necessary (```kD1D```). We addressed this point, as follows. First, we have added new experiments in Appendix E, to evaluate the impacts of using task-related (i.e., domain-relevant) stimuli. Empirical results show that using task-related stimuli results in 11 less distinct solutions, out of 350 generated solutions, and top-1 solution using task-related stimuli achieves a 6.99% lower miss ratio at 75th-percentile. Second, we have toned down Section 3.2. Instead of saying “Stimuli should be unbiased to the problem”, the text now says “Stimuli can be unbiased to the problem”.

**Cut-short Discussions with Reviewers.** Our discussion with Reviewer ```eCqA``` was cut short, due to the unfortunate OpenReview incident. We thank the reviewer for pointing out texts that might be confusing for other people. We hope that our last response to the reviewer is helpful, and we are happy to follow up.

---

### Meta-Review · Area_Chair_AWfq · 2026-01-04

**Summary:**

This work proposes MetaMuse, an LLM-based self-reflection framework, for automatic online system algorithm design. The three key components of MetaMuse are: 1) evaluating the solution diversity in the performance space using feedback measurements on n workload traces, 2) steering ideation through external stimuli (keywords from an English dictionary) with a carefully designed selection strategy powered by a performance prediction model (Gaussian Process Regression), and 3) constructing executable solutions using waypoint reasoning (a sequence of structured checkpoint-based reasoning steps). Experimental results show that MetaMuse can achieve promising performance on the cache replacement and online bin packing problems.

The reviewers initially had mixed scores (8,6,4,4) and raised concerns regarding the effectiveness and unbiasedness of external stimuli, waypoint reasoning, coding safeguards, computational overhead, problem selection, comparisons with other methods, experimental settings, and results. The authors have provided a concise rebuttal and revised the manuscript with additional discussion and analysis to address these concerns. The authors did not receive further responses from Reviewer 7Wx3, Reviewer wtEQ and Reviewer kD1D, while their discussion with Reviewer eCqA was cut short due to the OpenReview incident. None of the reviewers chose to raise their scores.

I have read the paper in detail and agree with the reviewers that this paper is well-written and easy to understand, the proposed MetaMuse method is novel and interesting, and the experimental results are strong on two real-world online system algorithm design problems. I believe many concerns (coding safeguards, computational overhead, comparisons with other methods, experimental settings, and results) have been adequately addressed by the rebuttal, but some of them (waypoint reasoning, problem chosen, code quality) might remain (partially) unresolved. A reasonable final score could be (8,6,6,4) for this paper (see discussion below).

Therefore, I consider this work to be borderline. I do recognize the positive impact it could have on the LLM-based algorithm design community, and I recommend acceptance if space permits.

**Reviewer Concerns:**

After the rebuttal, I believe many concerns raised by the reviewers have been properly addressed, though some remain (partially) unresolved:

**(Stimuli and) Waypoint Reasoning:** Reviewer kD1D, Reviewer wtEQ, and Reviewer eCqA have raised various issues regarding stimuli and waypoint reasoning. Having read the rebuttal, I think the remaining concern could be: how and why does waypoint reasoning effectively transform the selected stimuli, whether biased or not, into a novel and efficient algorithm (rather than a merely sophisticated random search)? I agree with the authors that providing a theoretical justification at this stage could be very challenging. However, a clearer description and discussion of the whole waypoint reasoning process, along with a well-designed analysis of the effect of each reasoning step, would be needed. At the end, Reviewer eCqA expressed dissatisfaction with the provided rebuttal and discussion, and I suspect Reviewers wtEQ and kD1D may also retain concerns on this point.

**Problem Chosen:** MetaMuse has only been tested on two online system algorithm design problems. While I acknowledge the authors' argument regarding the industrial value of these problems, I am uncertain why this prevents testing MetaMuse on additional problems to demonstrate its generality. Changing the title as suggested by Reviewer eCqA is another possible way to address this issue. It is encouraging to note in the authors' final summary comment that they are working to target more high-value, high-impact system algorithm problems.

**Code Quality:** I can also understand the concern regarding code quality raised by Reviewer eCqA. While it may be true that end-users of a commercial system care primarily about performance, I believe many algorithm researchers, including the ICLR audience in this subfield, would be interested in the quality and clarity of the generated code.

**Reviewer Scores:**

According to the rebuttal,  if the reviewers had been able to participate fully in the discussion, I believe:

**Reviewer 7Wx3** would maintain their positive score (8) to clearly support accepting this work.

**Reviewer eCqA** would not raise their score (6) due to the remaining concerns on waypoint reasoning, problem chosen, and code quality. On the other hand, I believe this reviewer would also not lower their score.

I am uncertain how **Reviewer kD1D** and **Reviewer wtEQ** would adjust their scores, as some of their concerns have been well addressed while others remain. A fair and reasonable estimate could be that one of them raises their score to 6, while the other keeps it unchanged (4).

Therefore, the reasonable final score for this paper could be (8, 6, 6, 4).

---

### Decision · Program_Chairs · 2026-01-26

Accept (Poster)